# Mesenchymal stem cells exert renoprotection via extracellular vesicle-mediated modulation of M2 macrophages and spleen-kidney network

Yuko Shimamura[1,6], Kazuhiro Furuhashi [1,6✉], Akihito Tanaka[1,6], Munetoshi Karasawa[1], Tomoya Nozaki[1], Shintaro Komatsu[1], Kenshi Watanabe[1], Asuka Shimizu[1], Shun Minatoguchi[1], Makoto Matsuyama [2], Yuriko Sawa[1], Naotake Tsuboi[3], Takuji Ishimoto[1], Hiroshi I. Suzuki [4,5] & Shoichi Maruyama [1]

Adipose-derived mesenchymal stem cells (ASCs) have shown therapeutic potentials against refractory diseases. However, the detailed therapeutic mechanisms remain unclear. Here, we report the therapeutic actions of human ASCs in nephritis, focusing on cellular dynamics and multi-organ networks. Intravenously-administered ASCs accumulated in spleen but not kidneys. Nevertheless, ASCs increased M2 macrophages and Tregs in kidneys and drove strong renoprotection. Splenectomy abolished these therapeutic effects. ASC-derived extracellular vesicles (EVs) were transferred to M2 macrophages, which entered the bloodstream from spleen. EVs induced the transcriptomic signatures of hyperpolarization and PGE2 stimulation in M2 macrophages and ameliorated glomerulonephritis. ASCs, ASC-derived EVs, and EV-transferred M2 macrophages enhanced Treg induction. These findings suggest that EV transfer from spleen-accumulated ASCs to M2 macrophages and subsequent modulation of renal immune-environment underlie the renoprotective effects of ASCs. Our results provide insights into the therapeutic actions of ASCs, focusing on EV-mediated modulation of macrophages and the spleen-kidney immune network.

[1] Department of Nephrology, Internal Medicine, Nagoya University Graduate School of Medicine, Showa-ku, Nagoya, Aichi, Japan. [2] Division of Molecular Genetics, Shigei Medical Research Institute, Minami-ku, Okayama, Japan. [3] Department of Nephrology, Fujita Health University School of Medicine, Toyoake, Aichi, Japan. [4] Division of Molecular Oncology, Center for Neurological Diseases and Cancer, Nagoya University Graduate School of Medicine, Showa-ku, Nagoya, Aichi, Japan. [5] Institute for Glyco-core Research (iGCORE), Nagoya University, Chikusa-ku, Nagoya, Aichi, Japan. [6] These authors contributed equally: Yuko Shimamura, Kazuhiro Furuhashi, Akihito Tanaka. ✉email: furu13@med.nagoya-u.ac.jp

Mesenchymal stem cells (MSCs) have shown beneficial effects in a broad range of animal disease models by promoting regeneration and/or enhancing immunoregulatory effects. MSCs have been clinically applied to various diseases, especially those that are difficult to treat with currently existing immunosuppressive drugs[1,2]. We started a clinical trial using adipose-derived MSCs (ASCs) for patients with refractory IgA nephritis (NCT04342325). However, the detailed therapeutic mechanisms remain unclear. Elucidating the therapeutic actions of human MSCs, which are used in clinical settings, rather than a rat or mouse MSCs, will allow us to provide higher and more stable therapeutic responses in clinical treatment. It is believed that the therapeutic effects of MSCs are achieved by paracrine signaling via secreted factors from MSCs[3]. Prostaglandin E2 (PGE2) derived from MSCs enhances IL-10 production from macrophages and increases the number of M2 macrophages[4–7]. Many other candidate molecules have been identified from analyses of secretory factors derived from MSCs, but none of them can fully explain their therapeutic effects. Recently, extracellular vesicles (EVs), including exosomes and microvesicles, have been intensively investigated as an in vivo delivery system that can transport multiple factors[8,9]. EVs contain various proteins, mRNAs, microRNAs, and mitochondria, and EV-transferred cells receive multiple factors at once, leading to the protection and regeneration of damaged organ[10–13]. Since the properties of EVs depend on the characteristics of the EVs produced[14,15], it is assumed that EVs produced from administered MSCs under inflammatory conditions in vivo acquire stronger anti-inflammatory functions than EVs in culture supernatants in vitro. However, only a few studies have investigated the characteristics of EVs derived from ASCs in vivo. Although it has been reported that intravenously administered MSCs accumulate in the lungs and spleen, the dynamics of MSCs in the body, including the organs in which they actually exert their therapeutic effects, are not fully understood. The aim of this study was to investigate the therapeutic actions of human ASCs by focusing on in vivo multiorgan cell dynamics, such as where and how the administered MSCs act, in a rat model of rapidly progressive glomerulonephritis. We report that ASCs enhance renoprotective effects by transferring EVs predominantly to M2 macrophages in the spleen, away from the inflamed kidney. Moreover, by tracking the EVs produced in vivo by the administered ASCs, we directly determined what modifications of M2 macrophages are caused by EVs secreted from ASCs in vivo. Overall, our study sheds light on the actions of ASCs, focusing on in vivo EV dynamics and the spleen-kidney immune network. Finally, our study maximizes the potential of cell therapies using ASCs in the clinical setting.

## Results

### Therapeutic potentials of human ASCs and BMMSCs for rapidly progressive glomerulonephritis.

Our previous report demonstrated the therapeutic potentials of rat ASC in an inflammatory model[4]. In this study, we determined the therapeutic effects of human ASCs or human BMMSCs in a rat glomerulonephritis model induced by the administration of a mouse monoclonal antibody (TF78) that binds to $\alpha4$(IV) non-collagenous (NC1) domains in the rat glomerular basement membrane (GBM) (Fig. 1a). In this model, administering pathogenic antibodies recapitulates the nephritis seen in human anti-GBM nephritis, and leads to rapidly progressive glomerulonephritis. First, we verified the human MSCs by the expression of cell-surface markers, including CD44, CD73, CD90, CD105, and the negative markers CD14, CD34, CD45, HLA-DR, and CD19 (Supplementary Fig. 1). Then, $2.0 \times 10^6$ ASCs or BMMSCs were administered intravenously to rats with anti-GBM nephritis

on days 0, 2, and 4 after intravenous injection of pathogenic TF78 antibody (Fig. 1a). Both ASCs and BMMSCs protected the kidney from renal damage, decreased serum creatinine and proteinuria (Fig. 1b, c), reduced glomerular damage scores, and reduced macrophage infiltration (Fig. 1d, e). When ASCs were compared with BMMSCs, ASCs showed stronger renal protective effects (Fig. 1b–e).

### Comprehensive flow cytometry analysis of renal leukocytes after MSC treatment.

To investigate whole renal leukocytes in glomerulonephritis after MSC administration, we analyzed leukocyte subsets in the kidney by flow cytometry (Fig. 2 and Supplementary Fig. 2a, b). In this flow panel, we identified whole leukocytes, T cells, CD4+ T cells, CD8+ T cells, Tregs, B cells, neutrophils, M1 macrophages, and M2 macrophages[16]. To determine whether the M1 and M2 macrophages identified by flow cytometry have functional anti-inflammatory and pro-inflammatory properties, we evaluated IL-10, IL-6, and CCR2 in cells purified by flow cytometry. M2 macrophages showed higher IL-10 expression and lower IL-6 and CCR2 expression than M1 macrophages, suggesting that this flow cytometry panel can clearly separate a subset of macrophages into anti-inflammatory and inflammatory macrophages (Supplementary Fig. 2c). As shown in a previous study using histological analysis[4], the number of macrophages and neutrophils peaked on day 3, and T cells increased from day 4 onward in this rat nephritis model. Myeloid lineage cells are involved in disease pathogenesis, and cells of the T-cell lineage accelerate the disease. To provide a comprehensive picture of the changes in a subset of leukocytes caused by the administration of MSCs, flow cytometry results were presented using t-distributed stochastic neighbor embedding (tSNE) (Fig. 2a, b). On day 3, both BMMSCs and ASCs decreased the number of M1 macrophages, but only ASCs reduced neutrophils and increased M2 macrophages (Fig. 2a, c, and Supplementary Fig. 3a). ASCs exerted a stronger effect on M2 cells from the myeloid lineage than BMMSCs did. Moreover, flow cytometry analysis revealed that ASCs induced Treg proliferation in the kidney in the early phase (day 3), leading to increased Treg frequency on day 7 (Fig. 2c, d). Furthermore, ASCs reduced the increased percentage of CD45 in living cells in the inflamed kidney during the early stage of nephritis (Supplementary Fig. 3a). The flow cytometry results suggest that not only ASCs inhibited leukocyte infiltration into the kidney, but also the anti-inflammatory effect of ASCs on myeloid lineage cells in the early stage of nephritis and the induction of Tregs by ASCs in late-stage nephritis converted the renal environment from an inflammatory state to an anti-inflammatory state, leading to amelioration of renal damage.

### ASC accumulation in the spleen leads to increased M2 macrophages and Tregs.

To determine where and how administered ASCs exert their therapeutic potential, we labeled ASCs with DiD (1,1'-Dioctadecyl-3,3,3',3'-Tetramethylindodicarbocyanine, 4-Chlorobenzenesulfonate Salt) cell membrane staining dye and performed microscopic observation the day after ASC administration in our glomerulonephritis model (Fig. 3a). Upon microscopic observation, we found that there were particles of different sizes, that is, large and small DiD-dye labeled particles. The large particles were ASCs themselves, while the small particles represented ASC membrane particles captured by other cells. In the spleen, liver, and lung, a large number of both ASC and ASC membrane particles was observed, with the highest number of ASCs in the lungs and the highest number of cells with ASC membrane components in the spleen (Fig. 3b). In the kidney, only a few DiD-positive cells were ASCs themselves. Rather, most

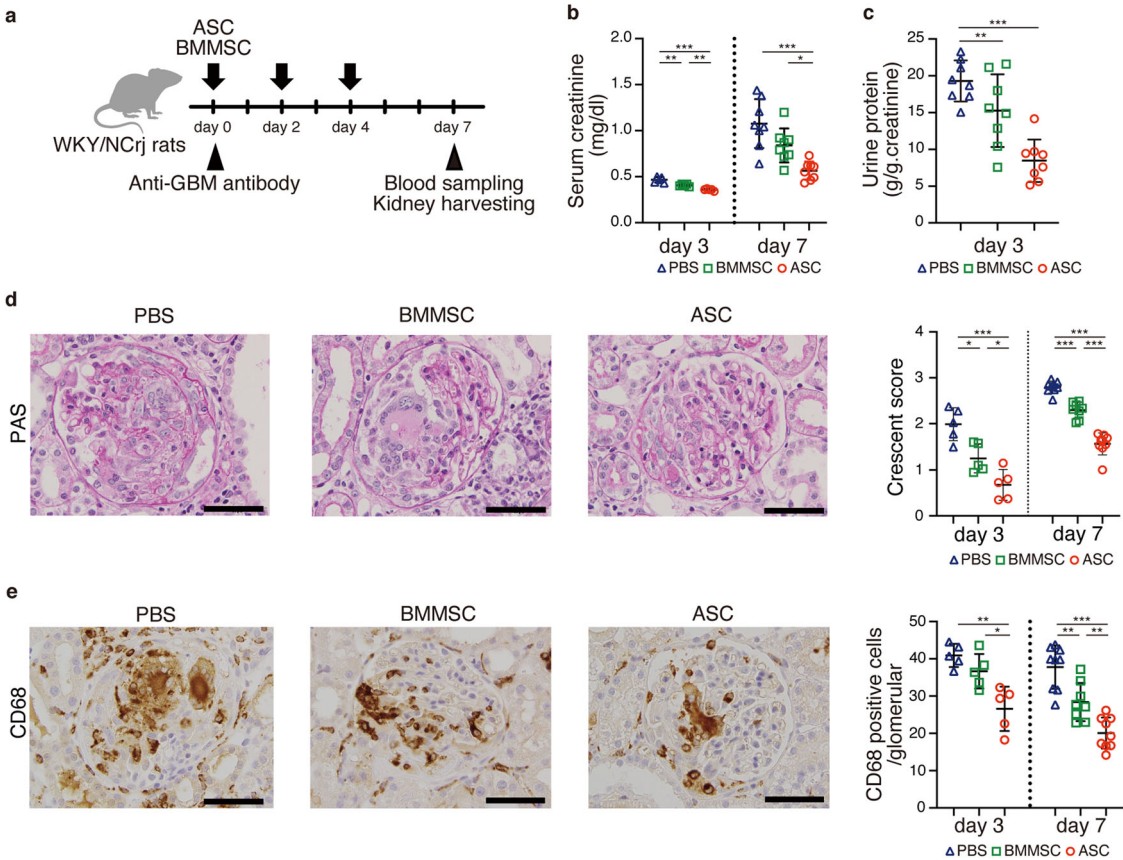

**Fig. 1 Human ASCs showed greater therapeutic effects on rapidly progressive glomerular nephritis than human BMMSCs. a** Experimental scheme depicting the induction of anti-GBM nephritis followed by intravenous ASC and BMMSC administration on days 0, 2, and 4. **b** Serum creatinine level on days 3 and 7 (day 3, $n = 5$ per group; day 7, $n = 8$ per group). **c** Urinary protein excretion on day 3 ($n = 8$ per group). **d** Representative images of kidney sections stained with Periodic Acid-Schiff on day 7 (left) and glomerular injury scoring based on crescent formation on days 3 and 7 (right).
**e** Representative images of kidney sections stained with anti-CD68 Ab on day 7 (left) and the number of CD68-positive cells in the glomerulus on days 3 and 7 (right). Scale bars, 50 μm. (day 3, $n = 5$ per group; day 7, $n = 8$–9 per group). All data are shown as means ± SD. *$p < 0.05$, **$p < 0.01$, ***$p < 0.001$, as determined by ANOVA. ASC adipose-derived MSC, BMMSC bone marrow-derived MSC.

were cells that captured ASC membrane components. The number of exogenous ASCs might be too small to explain the direct therapeutic effects on the kidney. BMMSCs also showed similar trends (Supplementary Fig. 4). To confirm whether the ASCs gathered in the spleen were functional cells, we evaluated the subset switch of leukocytes in the spleen by flow cytometry (Fig. 3c–f and Supplementary Fig. 3b). Comprehensive flow cytometry analysis showed that ASCs skewed the macrophage balance toward the anti-inflammatory side and increased B cells, CD4$^+$ T cells, and Tregs on day 3. On day 7, no change was observed in the ratio of M1 to M2 macrophages or Treg frequency in ASC- or BMMSC-treated rats. ASCs increased Treg proliferation, which was consistent with the flow cytometry results in the kidney, but at an earlier stage (day 3). When comparing BMMSCs and ASCs, the effect on myeloid cells was not different, but ASCs had a stronger effect on Treg induction and proliferation than BMMSCs (Fig. 3f). These findings indicate that ASC accumulation in the spleen shifts the balance of immune cells toward immune regulatory cells. Further, these results suggest that ASCs may regulate inflammation in the kidney by acting on the immune system in the spleen, which is located distant from the damaged organ.

**Impact of ASC accumulation in the spleen and lungs on treatment responses**. Next, we performed splenectomy to investigate the effects of ASC accumulation in the spleen. The

spleen was removed 1 week before nephritis induction, while the sham group received only a surgical incision. Evaluation of serum creatinine levels and histological scores revealed that splenectomy abolished the therapeutic effects of ASCs (Fig. 4a–c). Moreover, we assessed whether splenectomy affected changes in leukocyte subsets induced by ASC treatment, such as increased Tregs, decreased neutrophils, increased M2 macrophages, and decreased M1 macrophages using flow cytometry (Fig. 2c). Focusing on myeloid cells, even in the sham group without splenectomy, the effect of ASCs on myeloid lineage cells was abrogated simply by making a surgical incision (Fig. 4d, e, and Supplementary Fig. 3c). A previous study showed that creating skin wounds in steady-state mice enhanced the infiltration of myeloid lineage cells from the blood to the kidneys and had no effect on T cells[17]. Therefore, it was not possible to determine whether ASCs regulated macrophage subsets and neutrophils in this splenectomy model. However, splenectomy abolished ASC-induced Treg induction in the kidney (Fig. 4d, e). It enhanced the infiltration of CD45$^+$ leukocytes into the kidneys on day 3, but administration of ASCs reversed the leukocyte increase associated with splenectomy (Supplementary Fig. 3c).

To address whether the lungs play a role in the therapeutic effects of ASCs, we injected heparin to reduce ASC accumulation in the lungs (Supplementary Fig. 5a). Even when heparin administration reduced ASC accumulation in the lungs, no change was observed in the therapeutic effects of ASCs on

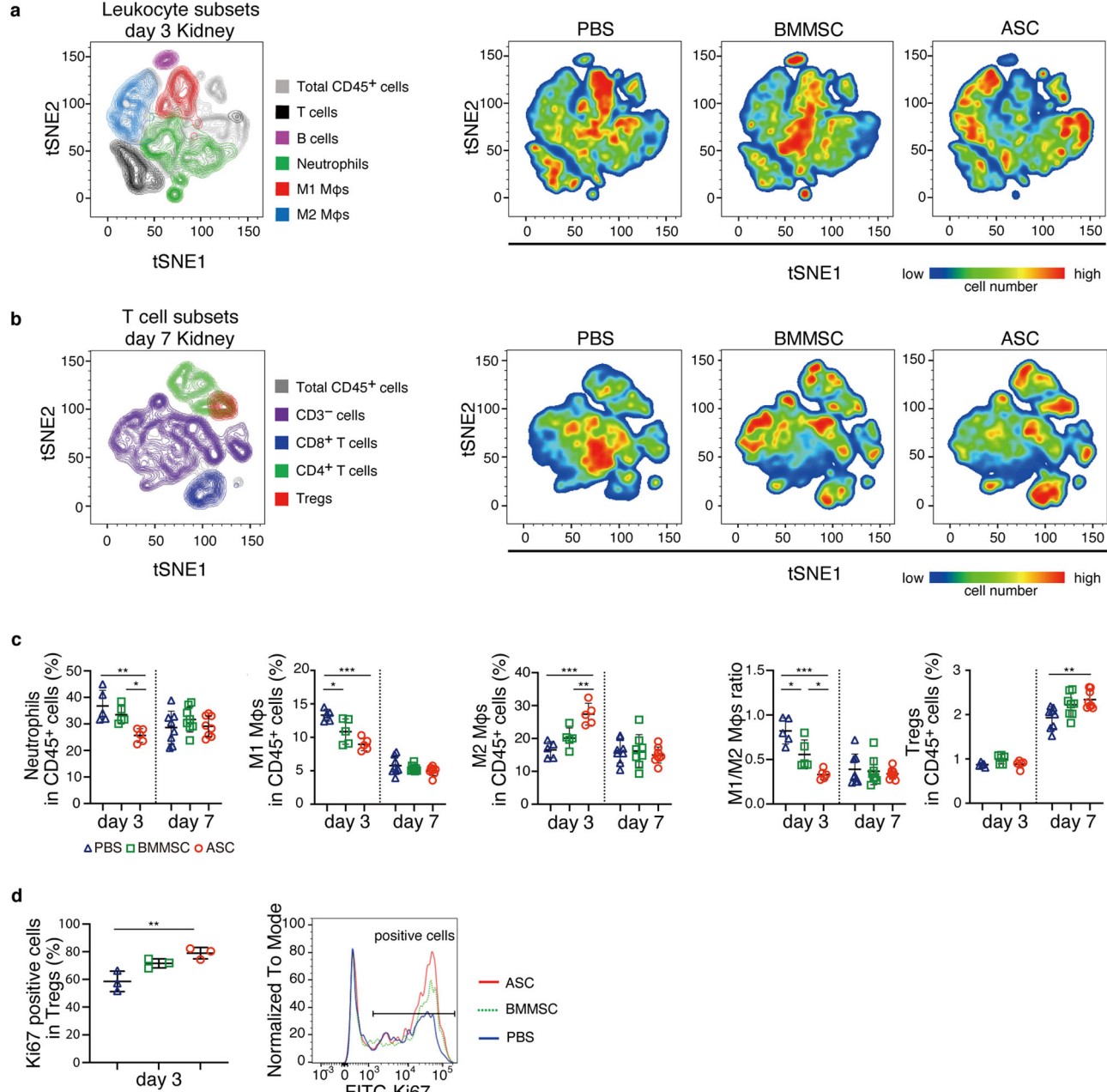

**Fig. 2 ASC treatment increased M2 macrophages and reduced neutrophils and M1 macrophages on day 3 and increased Tregs on day 7 in the kidney.**
**a**, **b** T-distributed stochastic neighbor embedding (tSNE) implementation for flow cytometry data visualization of leukocyte subsets and T-cell subsets. The data are presented as color-coded leukocyte population overlays (left) or cell number density color-coded (right) in the tSNE space. **a** Kidney samples on day 3 were divided into the following cell clusters: CD45+ cells, T cells, B cells, neutrophils, and M1 and M2 macrophages. **b** Kidney samples on day 7 were divided into the following clusters: CD45+ cells, non-CD3+ cells, CD8+ T cells, CD4+ T cells, and Tregs. **c** Frequency of renal leukocyte subsets in CD45+ cells analyzed by flow cytometry on days 3 and 7 (day 3, $n = 5$ per group; day 7, $n = 8$ per group). **d** The proportion of Ki67-positive cells in the renal Treg subset on day 3 ($n = 3$ per group). All data are shown as means ± SD. *$p \leq 0.05$, **$p \leq 0.01$, ***$p \leq 0.001$ as determined by ANOVA.

glomerulonephritis (Supplementary Fig. 5b). These results indicate that the spleen is a critical organ for the renoprotective effects of ASCs.

**Transfer of MSC-derived EVs to M2 macrophages in vivo.** We conducted studies using flow cytometry to quantify the number of DiD-dye labeled ASCs and leukocytes with DiD-dye labeled cell membranes in vivo. The DiD reagent can detect EVs, including exosomes and microvesicles, by labeling cell membranes[18]. In support of the histological result in Fig. 3b, flow cytometry analysis revealed that the number of DiD-

positive cells was higher in the spleen than in the kidney and that most of the DiD-positive cells in both the kidney and spleen were rat CD45+ cells, and not the administered ASCs themselves, which should be observed in the CD45−DiD+ fraction (Fig. 5a). These results indicate that either DiD-positive cell membranes were transferred to rat leukocytes as EVs or rat leukocytes phagocytosed DiD-positive ASCs as apoptotic bodies. To examine these two possibilities, we directly observed DiD location on rat CD45+ cells after cell sorting and in tissue sections stained with CD45 antibody (Fig. 5b, c). There were only a few ASCs, which were defined as CD45−DiD+ cells.

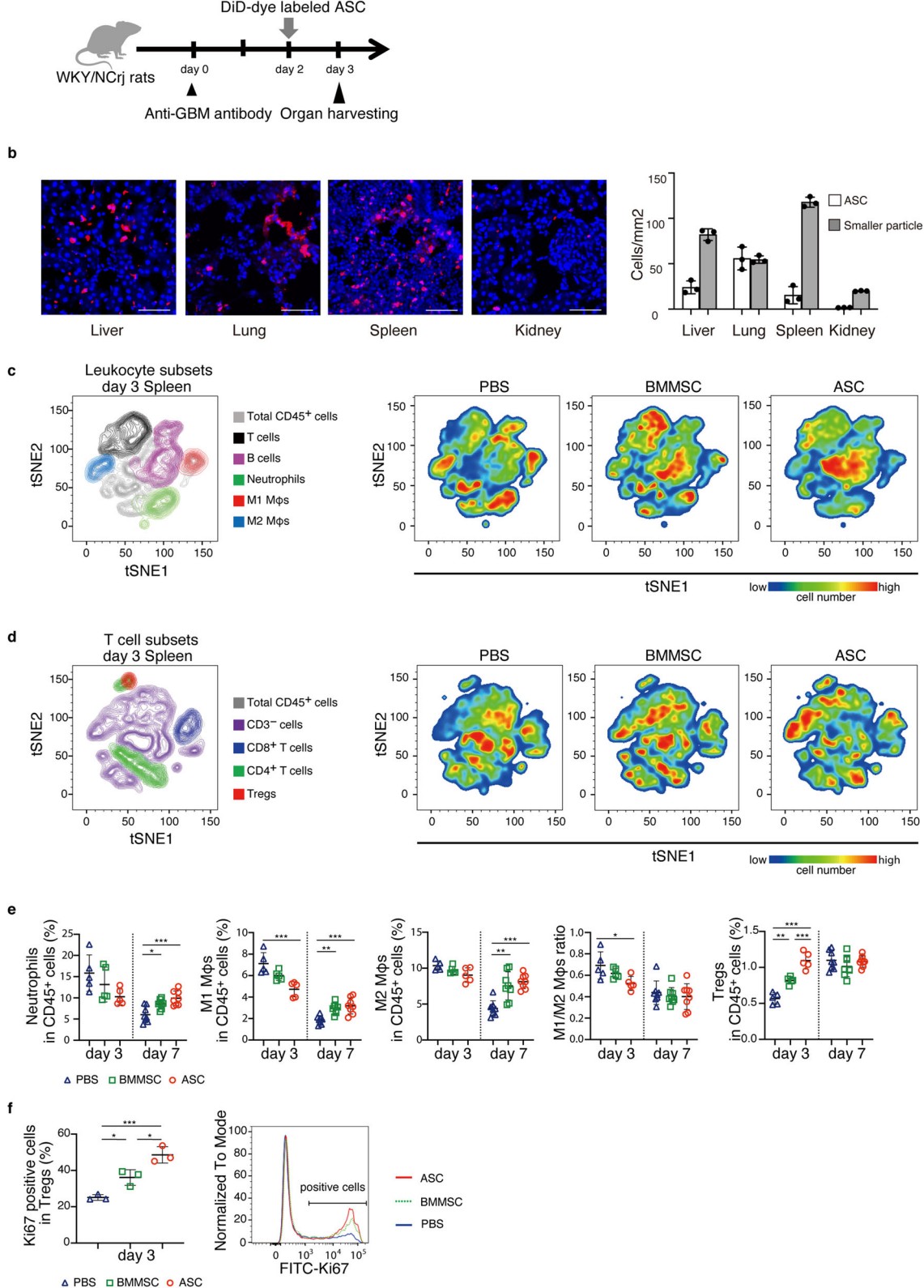

A small number of CD45$^+$ cells had DiD mainly in the cytoplasm, which suggested that leukocytes phagocytosed ASCs. In contrast, most of the DiD particles were observed on the surface of CD45$^+$ cells in both the spleen and kidney, suggesting that a large number of ASC-derived EVs were transferred to leukocytes.

Next, we found that DiD-positive MSC-derived EVs were mainly detected in M2 macrophages in the spleen and kidney by flow cytometry analysis. (Fig. 5d, e). Even as early as 4 h after ASCs administration, the majority of EVs were found in M2 macrophages, suggesting that ASCs-derived EVs were predominantly transferred to M2 macrophages (Supplementary Fig. 6).

**Fig. 3 Accumulated ASCs decreased M1 macrophages while increasing M2 macrophages and Tregs in the spleen. a** Experimental scheme for analyzing the distribution of administered ASCs in the body. DiD-dye labeled ASCs were administered to nephritis-induced WKY/NCrj rats on day 2. Tissues were harvested on day 3. **b** The number of DiD-positive cells with large DiD-dye particles and small DiD-dye particles in the liver, lung, spleen, and kidney is shown ($n = 3$ per group). Scale bars, 100 μm. **c**, **d** T-distributed stochastic neighbor embedding (tSNE) implementation for flow cytometry data visualization of leukocyte and T-cell subsets. The data are presented as color-coded leukocyte population overlays (left) or cell number density color-coded (right) in the tSNE space. **e** Frequency of splenic leukocyte subsets in CD45+ cells analyzed by flow cytometry on days 3 and 7 (day 3, $n = 5$ per group; day 7, $n = 8$ per group). **f** The population of Ki67-positive cells in splenic Treg subset on day 3 ($n = 3$ per group). All data are shown as means ± SD. *$p \leq 0.05$, **$p \leq 0.01$, ***$p \leq 0.001$ as determined by ANOVA.

The proportion of DiD-positive cell subsets in renal M2 macrophages was higher in the ASC group than in the BMMSC group (13.9 ± 0.4% and 10.3 ± 1.1%, respectively, $p = 0.0176$, Fig. 5f). In the spleen, there were no differences between the two groups (Fig. 5f).

**Transcriptional signatures of M2 macrophages via transfer of EVs secreted in vivo from ASCs**. To evaluate whether ASC-derived EVs affected the function of M2 macrophages, we performed quantitative PCR and showed that M2 macrophages with transferred EVs had enhanced IL-10 expression without significant changes in TGF-β, IL-6, and TNF-α levels (Fig. 6a). Next, we performed RNA sequencing (RNA-seq) to define the overall effects of ASC-derived EV transfer on M2 macrophages in vivo. We collected both DiD-positive (EV+) and DiD-negative (EV−) M2 macrophages from the spleen of the glomerulonephritis model using flow cytometry and performed RNA-seq analysis. M2 macrophages under physiological conditions (control) and glomerulonephritis without ASC treatment (GN) were also analyzed. We identified 809 differentially expressed genes (DEGs) among the four groups (FDR < 0.1) (Fig. 6b, c). Based on unsupervised clustering of these genes, gene set analysis using hallmark gene sets, and Gene Ontology (GO) biological process terms from the MSigDB database, we identified seven distinct DEG groups associated with distinct functions (Fig. 6b, d, e). Since DEG group 7 included granzyme B and possibly reflected contamination with NK or T cells, we focused on other DEG groups in subsequent analyses. Genes in DEG groups 1, 2, and 3 were upregulated by nephritis, and the expression of genes in DEG group 1 was further enhanced in EV+ samples relative to that in EV− samples (Fig. 6d). These gene sets were associated with several functions, such as secretion, exocytosis, glycolysis, and myeloid leukocyte activation (Fig. 6e and Supplementary Table 1). Notably, DEG group 1 was strongly related to secretion and exocytosis processes. The upregulated genes associated with secretion and exocytosis processes included (1) DEG group 1—*Hmgcr*, *Slc12a2*, *Pcsk1*, and *Llgl2*; (2) DEG group 2—*Anxa3* and *Itgam*; and (3) DEG group 3—*Fcgr2b* and *Tnfaip2* (Fig. 6c, e). On the other hand, the genes in DEG groups 4, 5, and 6 were downregulated by nephritis, and the expression of genes in DEG groups 4 and 6 were further suppressed in EV+ samples (Fig. 6d). These genes were enriched for IFN-γ, TNF-α, and NF-κB pathways, which are important for the induction of M1 macrophages, and included (1) DEG group 4—*Lpar6*, *Nr4a3*, *Hes1*, *Fosb*, *Phlda1*, and *Bcl2a1*; (2) DEG group 5—*Tnfsf13b* and *Serpinb8*; and (3) DEG group 6—*Cxcl10*, *Cxcl11*, *Ccrl2*, and *Zfp36* (Fig. 6c, e). DEG group 4 was also associated with other biological processes such as translation (Fig. 6e). A recent study has reported similar transcriptomic changes in M2 macrophages treated with succinate, including upregulation of secretion and exocytosis pathways and downregulation of genes that are preferentially expressed in M1 macrophages, which are indicative of hyperpolarization of M2 macrophages[19]. Therefore, these findings collectively suggest that M2 macrophages undergo hyperpolarization in nephritis and that EVs mediated a further phenotypic shift probably toward anti-inflammatory phenotypes. While M1

macrophages rely mainly on glycolysis, M2 macrophages are reported to be more dependent on mitochondrial oxidative phosphorylation. Upregulation of glycolysis-associated genes, such as *Plod1*, *Gusb*, and *Chst1*, in our data sets contradicts these findings (Fig. 6c, e); however, the importance of glycolysis in M2 macrophages has been recently suggested[20]. In addition, we observed that EV+ and EV− samples had very similar overall transcriptomes (Fig. 6b). This suggests that EVs may transiently contact even DiD-negative M2 macrophages in vivo and cause pervasive transcriptomic changes in anti-inflammatory macrophages.

While macrophages are largely divided into M1 and M2 macrophages, various external stimuli induce diverse transcriptional programs in macrophages[19]. Such complexity should be considered for a realistic consideration of EV effects because EVs can transfer various molecules. We utilized a previously reported database of transcriptomic effects of 28 different stimuli, including known M1 stimuli (IFN-γ), known M2 stimuli (IL-4), and prostaglandin (PGE), in human macrophages and examined similarities between these datasets and our datasets. To do so, we compared the expression changes of DEG groups identified in our datasets with those induced by 28 different stimuli[21]. As shown in Fig. 6f, DEG group 1, which was upregulated in nephritis and further increased in EV+ samples, showed the highest expression upon PGE2 stimulation. Importantly, this was highly consistent with our previous report that PGE2 produced by rat ASCs strongly induces the phenotypic conversion of macrophages into immunoregulatory cells[4]. Consistent with the hallmark gene set and GO analysis, DEG group 6, which was downregulated in nephritis and further decreased in EV+ samples, showed the highest expression upon IFN-γ stimulation. Collectively, these findings suggest that EVs secreted from the administered ASCs activate and/or hyperpolarize M2 macrophages, at least partially through PGE2 stimulation and downregulation of inflammatory pathways.

**Nephritis is ameliorated by administration of ASC-derived EVs**. We conducted an in vivo experiment to determine whether ASC-derived EVs have direct therapeutic potential without ASC administration. EVs secreted from cultured ASCs were administered to rats on days 0, 2, and 4 after nephritis induction (Fig. 7a). Electron microscopy confirmed that the pellets collected by ultracentrifugation from the culture supernatant contained EVs (Fig. 7b). Injecting ASC-derived EVs reduced the elevation of serum creatinine (Fig. 7c), but not proteinuria (Fig. 7d), and improved histological injury scores (Fig. 7e). These results indicate that ASC-derived EVs play an important role in the therapeutic effects of ASCs on glomerulonephritis, but the effect was not comparable to that of ASC administration itself.

**EVs-transferred M2 macrophage flux from the spleen to the bloodstream**. We performed time-lapse imaging by co-culturing ASCs with peritoneal macrophages in a culture dish. We captured the exact moment when ASCs released EVs and delivered EVs to macrophages (Fig. 7f and Supplementary Movie 1). This

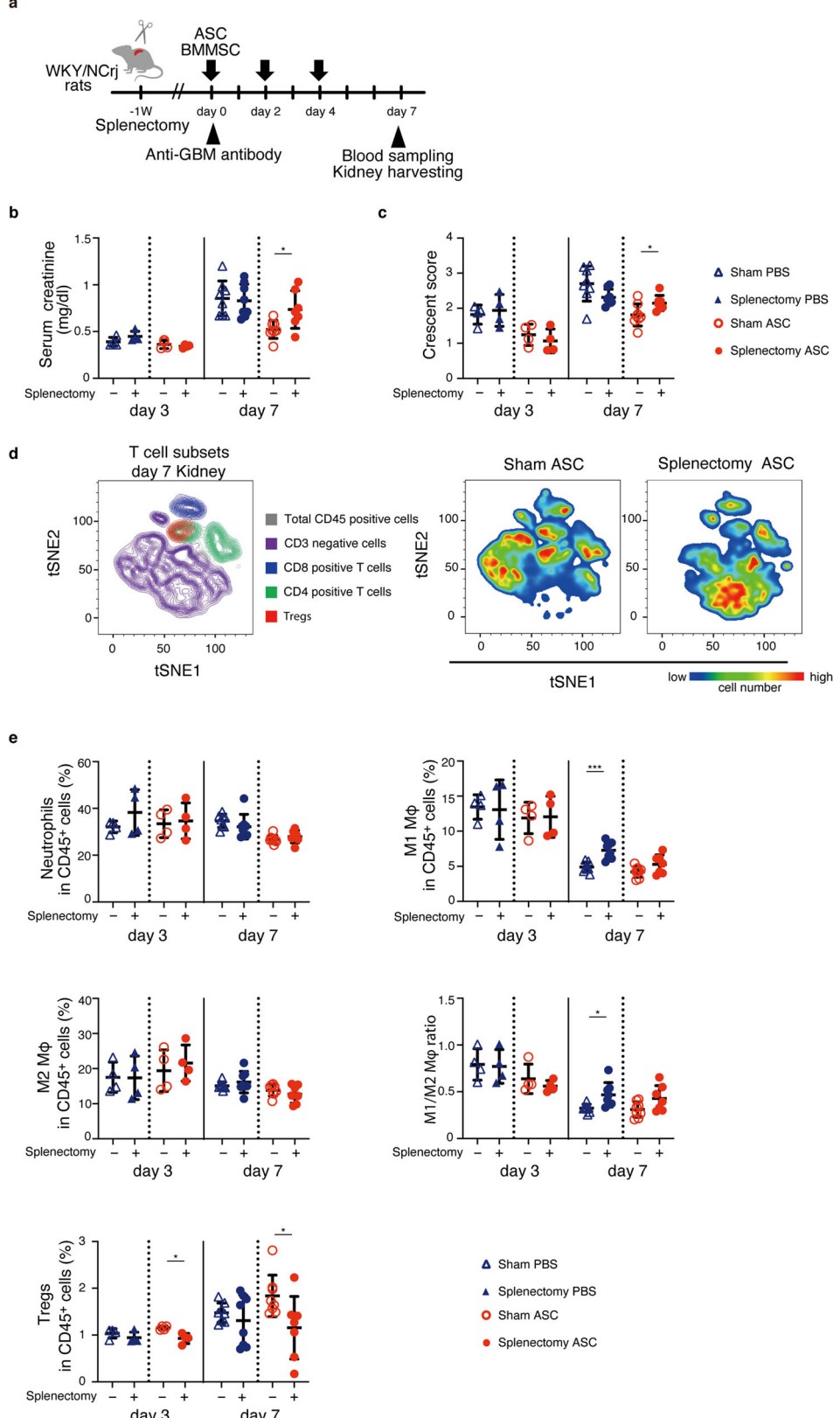

**Fig. 4 Splenectomy abolished the therapeutic potential of ASCs. a** Experimental scheme for analyzing the therapeutic potential of ASCs in rats treated with splenectomy. The rats underwent splenectomy 1 week before the nephrotoxic antibody injection. ASCs were administered on days 0, 2, and 4. **b**, **c** Serum creatinine level and histological crescent score on days 3 and 7 with and without splenectomy in each MSC treatment. **d** The tSNE plot shows the renal T-cell subset on day 3. **e** Frequency of renal leukocyte subsets in CD45+ cells analyzed by flow cytometry on days 3 and 7 (day 3, $n = 4$ per group; day 7, $n = 7$–8 per group). All data are shown as means ± SD. *$p \leq 0.05$, ***$p \leq 0.001$ as determined by Welch's $t$ test.

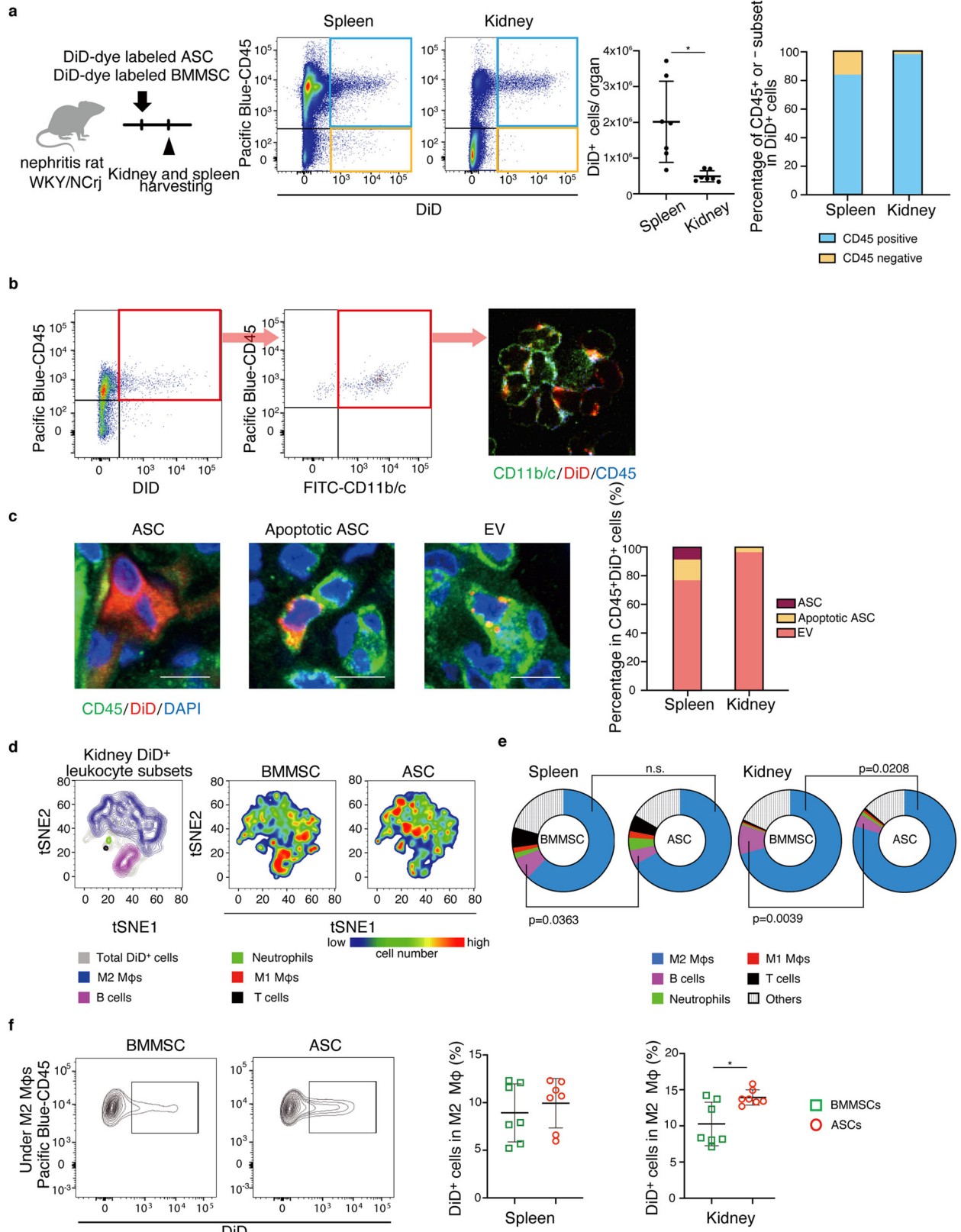

visualization directly showed that ASCs release EVs and pass them into M2 macrophages. Next, intravital microscopy was performed. At 48 h after the administration of DiD-dye labeled ASCs in nephritic rats, we labeled leukocytes by intravenous administration of FITC-labeled anti-CD45 antibody and labeled blood vessels by administration of rhodamine-labeled dextran.

Then, we observed the dynamics of EV-positive leukocytes in the spleen (Fig. 7g and Supplementary movie S2). We found that the splenic leukocytes, into which ASC-derived EVs were transferred, moved into the blood vessels and circulated through the bloodstream. We observed that ASCs increased M2 macrophages in the kidney on day 3 (Fig. 2c), while they slightly increased the ratio of

**Fig. 5 EVs from administered ASCs were predominantly transferred to M2 macrophages. a** Experimental scheme for analyzing the organ and cellular distribution of MSC membrane components by flow cytometry. The spleen and kidney were analyzed one day after the DiD-dye labeled ASC administration. The left graph shows the percentage of CD45-positive or negative subsets in DiD+ cells. The right graph shows the number of DiD+ cells in the spleen and kidney (n = 7 per group). **b** The confocal micrograph of CD11b/c+DiD+ cells sorted by flow cytometry. DiD particles were localized on the leukocyte membrane. **c** Representative images of ASC, Apoptotic ASC, and EV. Scale bars, 10 μm. The right graph shows the frequency of three DiD+ populations in the spleen and kidney (n = 3 per group). **d** tSNE plot from the flow cytometry analysis shows DiD+ leukocyte subsets. **e** Pie charts show the proportion of each subset in DiD+ leukocytes in the spleen and kidney (n = 7 per group). **f** Representative images and results of flow cytometry analysis show DiD+ cell frequency in renal M2 macrophages after administration of DiD-dye labeled ASCs or BMMSCs into nephritis rats (n = 7). All data are shown as means ± SD. *p ≤ 0.05 as determined by Welch's t test.

M1 to M2 macrophages in the spleen on day 3 (Fig. 3e). Therefore, rapid entry of M2 macrophages from the spleen into the bloodstream may contribute to the early stage (day 3) increase in M2 macrophages in the kidney.

**Direct induction of Tregs by ASCs, ASCs-derived EVs, and EV-positive M2 macrophages.** Finally, to validate our findings in vivo, we conducted in vitro experiments focusing on Treg induction. We determined whether ASCs, ASC-derived EVs, and ASC-EV-transferred M2 macrophages directly induced Tregs from T cells in vitro. Since it was unclear whether ASCs acted directly on T cells or acted via other cells, we first co-cultured bone marrow cells with ASCs and evaluated the bone marrow cells for Treg induction. ASCs induced Tregs during co-culturing with bone marrow cells (Fig. 8a). Next, we co-cultured ASCs with CD4+ T cells purified by flow cytometry to demonstrate the direct interaction between them. ASCs induced Tregs from purified CD4+ T cells (Fig. 8b). Moreover, ASCs-derived EVs alone was able to induce Tregs from purified CD4+ T cells in co-culture (Fig. 8c). We showed that the transfer of ASC-derived EVs enhanced the ability of M2 macrophages to induce Tregs (Fig. 8d). The presence of ASC-derived EVs and M2 macrophages carrying EVs induced Tregs from CD4+ T cells, indicating that Tregs can be induced even though there are few ASCs in the inflamed kidney.

**Discussion**
We investigated the therapeutic actions of ASCs, focusing on EV-mediated modulation of macrophages and the spleen-kidney immune network, which may maximize the potential of cell therapies in the clinical setting. The rat model of rapidly progressive glomerular nephritis was chosen for the following reasons: it is comparable to human anti-basement membrane antibody nephritis; the induced nephritis rapidly progresses; the renal prognosis is quite poor with existing treatment, and immune cells are deeply involved in the disease pathogenesis[22]. We showed that human ASCs exert remarkable therapeutic effects compared to human BMMSCs. Flow cytometry analyses of leukocytes revealed that ASCs shift overall leukocyte characteristics from an inflammatory to an anti-inflammatory state. Intravital imaging and flow cytometry allowed us to elucidate the dynamics of ASCs and EVs in vivo. Our results clearly demonstrate that ASC-derived EVs are mainly transferred to M2 macrophages and that ASC-derived EVs activate M2 macrophages via PGE2 in the spleen, which then enter the bloodstream, leading to an increase in Tregs and M2 macrophages, while decreasing M1 macrophages in the kidney (Fig. 9). This is the first study to clarify the therapeutic effect of ASCs by focusing on the interrelationship between ASCs, EVs, PGE2, the spleen, and the damaged kidney.

One of the findings of the present study is that the spleen is the key organ for the therapeutic effects of ASCs in a nephritis model. The spleen is a reservoir of immune cells and contains both cells that promote inflammation and cells that inhibit inflammation. In fact, there have been reports of cerebral infarction improvement with splenectomy[23], while pulmonary damage worsens with splenectomy, suggesting that the role of the spleen may depend on the situation[24]. Although there are few reports showing that MSCs act on splenic cells and improve injuries in distant organs, Badner et al. reported that splenectomy abrogated the therapeutic effects of MSCs on traumatic spinal cord injury[25]. Our microscopic observations showed that a large number of fluorescent dye-positive cells were found in the spleen, while only a small number of labeled cells were observed in the kidney. To evaluate whether ASCs improve nephritis by acting on splenic cells, the spleen was removed one week before the administration of ASCs. This canceled the renoprotective effect and Treg induction by ASCs, indicating that the interaction of ASCs with immune cells in the spleen is essential for the therapeutic effects against kidney injury.

Another important finding of our study is that EVs secreted from ASCs in vivo are transferred predominantly to M2 macrophages and modulate M2 macrophage function via PGE2 stimulation-like transcriptome changes. To examine the exact number of MSCs in the kidney and spleen, flow cytometry analysis was performed. Surprisingly, there were few labeled MSCs in the kidney. Instead, the cells that were positive for the fluorescent dye were mostly leukocytes. In contrast, fluorescent MSCs and fluorescent leukocytes were observed in the spleen. Since DiD-dye is a cell membrane-labeling reagent, there were two possibilities: one was that ASCs were phagocytosed by leukocytes, and the other was that ASC-derived EVs were transferred to leukocytes. To examine these possibilities, we directly observed the DiD-dye distribution in the cells using high-resolution microscopy. If the ASCs are phagocytosed, membrane-labeled dyes should be observed in the leukocyte cytoplasm, and if plasma membrane components, such as EVs are transported, labeled dyes should be found on the plasma membrane. We found that most of the membrane components were distributed on the leukocyte cell membrane, suggesting that the membrane components were transferred to leukocytes as EVs. EVs have been reported to act on various immune cells, such as T cells, B cells, dendritic cells, and macrophages[9,26–28]. To identify which cell subset the ASC-derived EVs act directly on, we tracked the membrane components of the administered ASCs. This enabled us to clearly determine that EV-positive cells were M2 macrophages in the spleen and kidney using flow cytometry, and allowed us to successfully collect the target cells for RNA-seq analysis. To the best of our knowledge, this is the first transcriptome study to analyze phenotypic changes in M2 macrophages induced by EVs secreted in vivo by ASCs. Considering the diversity of M2 macrophages, we investigated the EVs stimuli that were comparable to the 28 stimuli by comparing the transcriptome changes in M2 macrophages induced by the 28 stimuli. These results suggest that EV transfer facilitates hyperpolarization of M2 macrophages in nephritis conditions further in the M2 direction possibly via PGE2 stimulation.

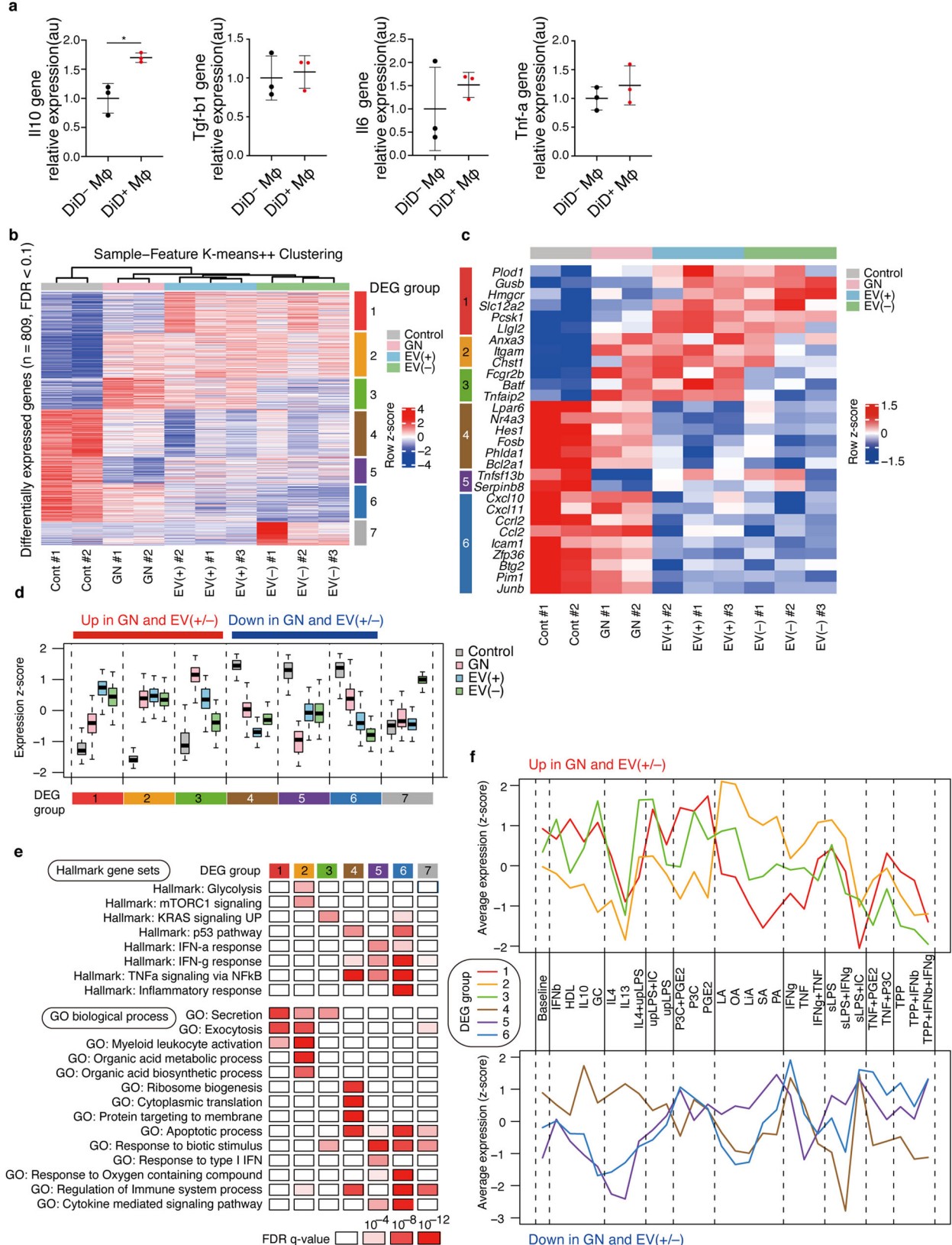

Furthermore, we analyzed the anti-inflammatory functions of EVs in vivo. Cultured ASC-derived EVs were administered to a rat model of nephritis. EVs protected rats from kidney injury, which is consistent with previous studies that have shown that MSCs-derived EVs protected from organ injuries[29,30]. The therapeutic potential of EVs was weaker than that of ASC administration itself. Inflammatory stimulation with interferon-gamma has been reported to enhance the anti-inflammatory function of MSCs or MSC-derived EVs[31,32]. We have elucidated certain potential mechanisms of MSCs for renoprotection, but these do not sufficiently explain all actions. To examine whether the EVs secreted in vitro are the same as those secreted in vivo

**Fig. 6 Transcriptomic effects of ASC-derived EVs on splenic M2 macrophages in vivo. a** Comparison of gene expression levels between DiD+ M2 macrophages and DiD− M2 macrophages (n = 3 per group). All data are shown as mean ± SD. *p ≤ 0.05 as determined by Welch's t test. **b** K-means ++ clustering analysis of DEGs among four sample groups in RNA-seq datasets (FDR < 0.1). Color scales are normalized along each row. **c** Gene expression patterns of representative genes are shown in **b**. **d** Box plots showing expression changes of DEG groups 1–7 in control, GN, EV (+), and EV (−) samples. **e** Summary of representative hallmark gene sets and gene ontology (GO) biological process terms associated with each DEG group with corresponding FDR q values, determined by MSigDB gene set overlap analysis. **f** Expression changes of genes in DEG groups in human macrophages stimulated with 28 different stimuli. Datasets in a previous report were reanalyzed.

after inflammatory stimuli, further studies are needed. There have been no reports on the analysis of EVs produced in vivo by MSCs administration into the body. Our results obtained from analyzing the changes in M2 macrophages caused by MSCs- derived EVs in vivo using RNA-seq and exploring the stimulating factors for these changes could contribute to further investigations on MSCs-derived EVs.

Intravital microscopy revealed that the EV-positive leukocytes in the spleen moved into the splenic blood vessels and entered the bloodstream, suggesting that M2 macrophages whose activity is enhanced by EVs are released from the spleen, reaching the inflamed kidney via the bloodstream. To the best of our knowledge, this is the first study to directly track the dynamics and function of EVs secreted by ASCs in vivo. Taken together, we successfully demonstrated the in vivo dynamics of ASCs and ASC-derived EVs, and clarified the importance of M2 macrophages that receive EV transfer in a model of nephritis (Fig. 9).

In vitro experiments were also conducted to examine the anti-inflammatory effects of EVs. These results show that ASC-related Treg induction can be achieved by ASCs themselves, ASCs-derived EVs, and EV-transferred M2 macrophages. The presence of ASC-derived EVs and M2 macrophages carrying EVs can induce Tregs from CD4 T cells, indicating that Tregs can be induced in the kidney, even though there are few ASCs in the inflamed kidney.

One of the limitations of this study is that human ASCs were administered to rats. The interaction of human ASCs with rat immune cells may not be the same as reactions with human immune cells. To elucidate the actual therapeutic actions of human ASCs, investigations using human samples obtained from clinical studies are needed. We have recently begun clinical trials using ASCs for patients with refractory IgA nephritis (NCT04342325). We plan to analyze human samples to validate the potential mechanisms that we have observed in this study.

In summary, our study indicates that EVs released from ASCs are transferred predominantly to M2 macrophages in the spleen, which then enters the bloodstream and reach the kidneys. There, the EV-transferred M2 macrophages modify leukocytes in the kidney from an inflammatory to anti-inflammatory state, resulting in renal protection. Furthermore, an analysis of the in vivo dynamics of ASCs revealed that ASCs are almost absent in the kidney, while M2 macrophages with ASC-derived EVs were found in the kidney. We also demonstrate that EVs enhanced the immunoregulatory functions of M2 macrophages via PGE2, and that inflammation in the kidney is regulated by the spleen, which is far from the kidney (Fig. 9). Although further studies are needed, the present study shows that ASC treatment has great potential as a therapeutic option for inflammatory kidney diseases and provides insights into therapeutic avenues for ASCs regarding where and how ASCs exert their therapeutic effects in vivo.

## Methods

**Animals**. WKY/NCrj rats were purchased from Charles River, Inc. (Yokohama, Japan). All rats were maintained on a 12-hour light/dark cycle with free access to standard diet and water, in accordance with the National Institutes of Health Guide for the Care and Use of Laboratory Animals. All animal experimental protocols were conducted in accordance with the guidelines of the Institutional Animal Ethics Committee of Nagoya University Graduate School of Medicine (Approval number 20377). Female 7–12-week-old rats were used for the nephritis model. The ages of the rats were matched for each MSC treatment experiment.

**Isolation and culture of human ASCs**. After obtaining informed consent to harvest mesenchymal stem cells for research purposes from the adipose discarded around the donor kidney in kidney transplantation, mesenchymal stem cells were cultured. All experiments using human tissue samples were approved by the ethical committee at the Nagoya University Medical School (approval number 2005-0347-5) and adhered to the guidelines of the Declaration of Helsinki. ASCs were isolated and cultured as previously described[33–35]. Briefly, the specimens were cut into 2-mm cubic pieces. Adipose tissue was digested in Hank's balanced salt solution containing 1 mg/mL collagenase type I (Worthington Biochemical Corporation, Lakewood, NJ, USA) for 1 h at 37 °C. Digested tissue was passed through a 100-μm pore filter to remove undigested debris. The obtained stromal vascular fraction were cultured in the medium that contained 3:2 mixture of Dulbecco's modified Eagle's medium (DMEM; Nissui Pharmaceutical Co. Ltd,Tokyo, Japan) and MCDB 201 medium (Sigma-Aldrich, St Louis, MO, USA), supplemented with 1 ng/ml linoleic acid-albumin (Sigma-Aldrich), one hundredth volume Insulin, Transferrin, Selenium (ITS) supplement (Sigma-Aldrich), 0.1 mM ascorbic acid phosphate ester magnesium salt (Wako Pure Chemical Industries, Osaka, Japan), 50 U/ml penicillin, 50 mg/ml streptomycin (Meiji Seika Ltd, Tokyo, Japan) and 2% fetal bovine serum (Sigma-Aldrich).

**Induction of rapidly progressive glomerulonephritis and cell therapy with MSCs**. Rapidly progressive glomerulonephritis was induced in WKY/NCrj rats by administering 100 μg/rat of the mouse monoclonal IgG clone TF78, which specifically binds to a4(IV) NC1of the rat glomerular basement membrane[4]. MSCs were administered intravenously on days 0, 2, and 4 after the induction of nephritis. Proteinuria and serum creatinine were evaluated from urine samples obtained on day 3 and blood samples obtained on days 3 and 7 after nephritis induction to evaluate renal function. Urine protein was measured by BML, Inc. (Tokyo, Japan), while serum creatinine was measured by SRL, Inc. (Tokyo, Japan). To histologically assess the disease severity, semi-quantitative analysis of glomerular crescent formation was performed on paraffin-embedded tissue sections using the periodic acid–Schiff staining method. The percentage of crescent occupying the area in each glomerulus was estimated and assigned for scoring as described[4]: 0, no crescent; 1, 0–25%; 2, 25–50%; 3, 50–75%; and 4, 75–100% crescent occupation. To evaluate macrophage infiltration, formalin-fixed kidneys were stained using mouse anti-rat CD68 monoclonal IgG1 (clone ED-1; BMA Biomedicals, Augst, Switzerland) as a marker for infiltrating macrophages or dendritic cells. For each animal, crescent formation and the number of stained cells were evaluated in >30 glomeruli per renal cross-section.

**Tracking intravenously-administered ASCs in rat organ tissues by microscopy**. The cell membranes of MSCs were labeled with Vibrant DiD-dye (Thermo Fisher Scientific, MA, USA) to evaluate cell dynamics in vivo and in vitro. ASCs or BMMSCs (2 × 10^6 cells) labeled with DiD dye were administrated into the nephritis rats via the tail vein on day 2, and tissue samples were extracted on day 3 and fixed with 1% paraformaldehyde buffer. The tissue sections stained with DAPI (Vector Laboratories, CA, USA) were observed using a Nikon TiE-A1R laser scanning confocal microscope (Nikon, Tokyo, Japan). The number of nuclei overlapping with DiD-dye was evaluated as DiD+ cells in the liver, lung, spleen, and kidney using Imaris (Biplane, Belfast, UK). Large particles with an estimated size of 10 μm were counted using Imaris software, which were recognized as ASCs. Smaller particles were also counted[36]. Next, we evaluated the location of DiD fluorescence in DiD-positive cells in the spleen and kidney by staining with the FITC-anti-rat CD45 antibody (Biolegend, San Diego, CA, USA) followed by fluorescein/Oregon Green polyclonal antibody labeled with Alexa Fluor 488 (Thermo Fisher Scientific, MA, USA) and DAPI. More than thirty DiD particle-positive cells were evaluated for their DiD fragment distribution. CD45−DiD+ cells were defined as ASCs. CD45+DiD+ cells, in which DiD particles occupied more than 1/3 of the cytoplasm area, were considered as leukocytes phagocytosing ASCs. CD45+ cells with DiD particles on the cell membranes were identified as EV-transferred leukocytes. Antibodies and dyes used in histological staining were listed in Supplementary Table 2.

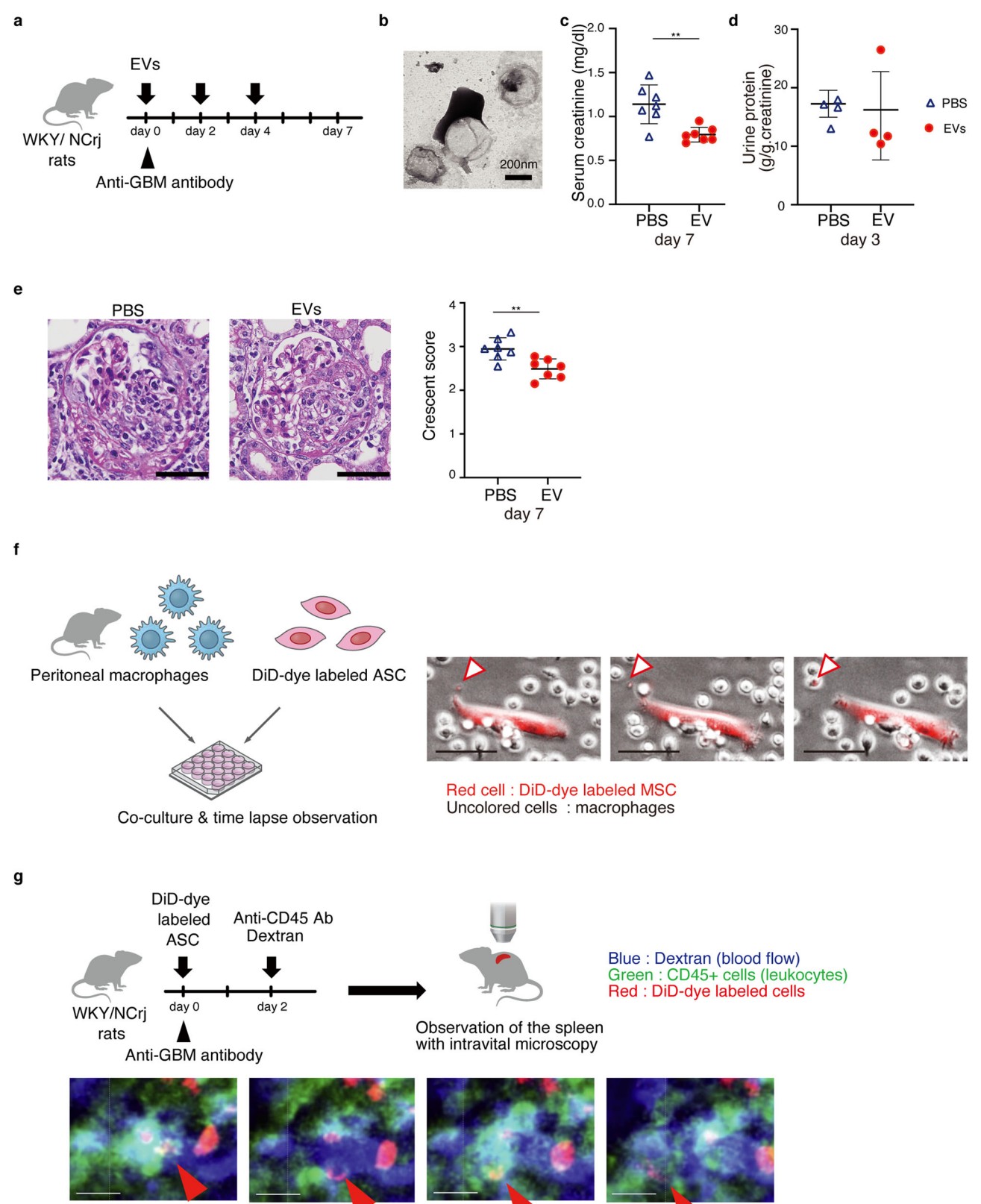

**Flow cytometry analysis of kidney and spleen samples**. For flow cytometric analysis of leukocytes in the kidney and spleen, kidneys and spleen were harvested after peripheral blood was eliminated using the blood withdrawal procedure. Therefore, circulating monocytes were not included in the present analysis. Tissue samples were cut into small fragments using a razor and suspended in media containing 1 mg/mL collagenase type I (Worthington Biochemical Corporation, Lakewood, NJ, USA) and 1 mg/mL DNase I (Merck, Darmstadt, Germany).

Samples in the dissociation solution were incubated at 37 °C and gently agitated for 30 min. The tissue fragments were then passed several times through an 18-gauge needle using 1 mL syringes and filtered through a 100 μm strainer. The collagenase reaction was stopped by adding PBS with 5 mM EDTA (Thermo Fisher Scientific, MA, USA). The isolated cells were pelleted at $400 \times g$ at 4 °C. The red blood cells in the samples were depleted with BD Pharm Lyse (BD, San Jose, CA, USA). The cells were blocked with anti-CD32 antibody (1 in 50 BD, 550271) and stained with the

**Fig. 7 EVs have therapeutic potential for anti-glomerular nephritis. a** Experimental scheme to determine therapeutic potentials of EVs for anti-GBM nephritis. EVs were administered intravenously on days 0, 2, and 4. **b** Electron micrograph of EVs. **c** The serum creatinine level on day 7 after EV treatment ($n = 7$ per group). **d** The urine protein excretion on day 3 ($n = 4$ per group). **e** Representative images of kidney sections stained with Periodic Acid-Schiff on day 7. Scale bars, 50 μm. The glomerular injury was evaluated using crescent score ($n = 7$ per group). **f** Experimental scheme of the co-culture of peritoneal macrophage and DiD-dye labeled ASCs. Time-lapse imaging show that EVs generated from the DiD-dye labeled ASCs are captured by macrophages. Scale bars, 50 μm. **g** Experimental scheme for observing dynamics of EV-transferred leukocytes using intravital microscopy. Time-lapse imaging of the spleen shows that the DiD$^+$ leukocytes move into the bloodstream. Scale bars, 10 μm. **p ≤ 0.01 as determined by Welch's t-test.

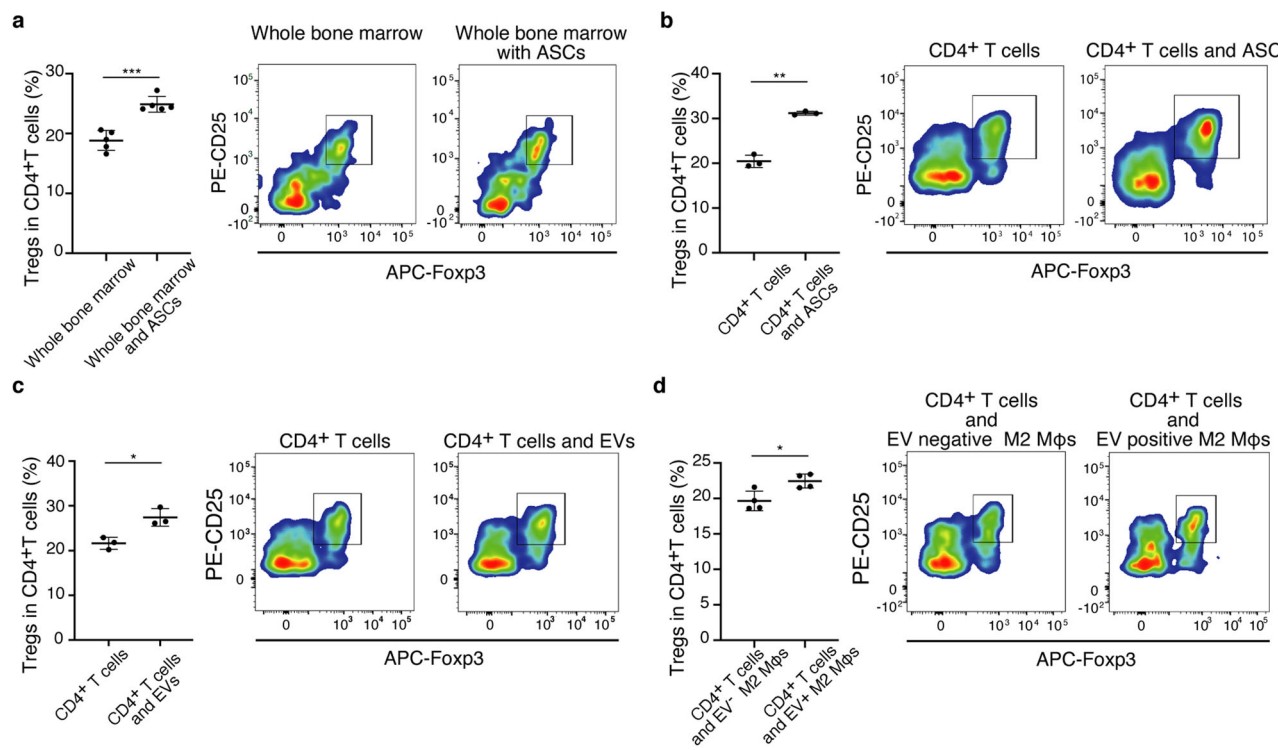

**Fig. 8 EVs and EV-transferred M2 macrophages can induce Tregs in vitro. a** The co-culture of whole bone marrow cells and ASCs. ASCs enhanced Treg induction in CD4$^+$ T cells ($n = 5$ per group). **b** The co-culture of CD4$^+$ T cells sorted from bone marrow and ASCs ($n = 3$ per group), **c** the co-culture of CD4$^+$ T cells sorted from bone marrow and EVs derived from cultured ASCs ($n = 3$ per group), and **d** the co-culture of CD4$^+$ T cells sorted from bone marrow and M2 macrophages with EVs ($n = 4$ per group) induced Tregs in CD4$^+$ T cells. All data are shown as means ± SD. *$p ≤ 0.05$, **$p ≤ 0.01$, and **$p ≤ 0.001$ as determined by Welch's t test.

labeled monoclonal antibodies listed in Supplementary Table 3, B–D. Leukocyte subsets were gated as shown in Fig. S2, and analyzed using FlowJo software (v10) (BD Bioscience, Tokyo, Japan). For flow cytometric sorting, cells were sorted using a FACS SORP Aria II (BD Bioscience, Tokyo, Japan). Cytospin4 Cytocentrifuge (Thermo Fisher Scientific, MA, USA) was used to observe the sorted cells under the microscope. For quantitative RT-PCR analysis and RNA-Seq analysis, cells were sorted using a FACS SORP Aria II after isolating cells with biotin anti-rat CD11b/c antibody (Biolegend, San Diego, CA, USA) and Streptavidin Particles Plus (BD, San Jose, CA, USA).

**Quantitative RT-PCR**. RNA from sorted cells (M1 macrophages, M2 macrophages, and DiD$^+$/DiD$^-$ M2 macrophages) was extracted using RNeasy Plus Micro Kits (Qiagen, Hilden, Germany). Reverse transcription was performed using PrimeScript RT Master Mix Perfect Real-Time (Takara Bio, Shiga, Japan). Gene expression was determined using a StepOnePlus Real-Time PCR System (Thermo Fisher Scientific, MA, USA). The primers were synthesized by Takara Bio (sequences listed in Supplementary Table 4).

**RNA-seq analysis**. For RNA-seq analysis, rat splenic M2 macrophages from four groups were collected by flow cytometry: control (physiological condition), GN (glomerulonephritis without ASC treatment), EVs$^+$ (EVs transferred during glomerulonephritis), and EVs$^-$ (EVs untransferred during glomerulonephritis). RNA-seq experiments were performed in two or three biological replicates. RNA was extracted using an RNeasy Plus Micro Kit (Qiagen). Libraries for RNA-seq were prepared using SMART-Seq v4 Ultra Low Input RNA Kits for Sequencing (Clontech, Tokyo, Japan) and Nextera XT DNA Library Prep Kits (Illumina, San Diego, CA). RNA libraries were sequenced using paired-end sequencing using a

NovaSeq 6000 instrument (Illumina). The sequencing reads were aligned to the rn6 reference genome using STAR (v2.5.3)[37]. Reads in each refSeq gene were counted with HTSeq (v0.6.0) using the intersection-strict model[38]. The edgeR package in R was used to identify the DEGs with an FDR threshold of 0.1[39]. After filtering out genes with a maximum count per million less than 1 across all samples, the trimmed mean of M-values normalization method, and generalized linear models were used to compare gene expression data. DEGs among the four groups (FDR < 0.1) were grouped into seven clusters using K-means ++ clustering. Gene set overlap analysis was performed using the MSigDB database using hallmark gene sets and GO biological process gene sets (https://www.gseamsigdb. org/gsea/msigdb/index.jsp). Transcriptome datasets with 28 different stimuli were previously described[21]. The average expression levels of genes in each DEG group were determined for the 28 different stimuli.

**EV isolation**. The supernatant from $3$–$4 × 10^6$ ASCs was ultracentrifuged using a W32Ti rotor (L-80XP; Beckman Coulter, Brea, CA, USA) at $100,000 × g$ for 70 min to pellet the exosomes. EVs were observed using a JEM-1400 electron microscope (JEOL, Tokyo, Japan). For the treatment of nephritis rats, EVs from $3$–$4 × 10^6$ ASCs were administrated intravenously on days 0, 2, and 4. Protein content was evaluated using Pierce BCA Protein Assay Kits (Thermo Fisher Scientific, MA, USA, 23225). The yield of EVs from $3$–$4 × 10^6$ ASCs was $318.7 ± 50.6$ μg.

**Time-lapse recording of co-cultured DiD-labeled ASCs and peritoneal macrophages**. Peritoneal macrophages were obtained from peritoneal lavage of WKY rats by intraperitoneal injection of 50 mL sterile saline. The abdomen was gently massaged before the lavage collection. The lavage fluid was centrifuged at $400 × g$ for 30 min. Mononuclear cells in the pellet were isolated using Histopaque-1083

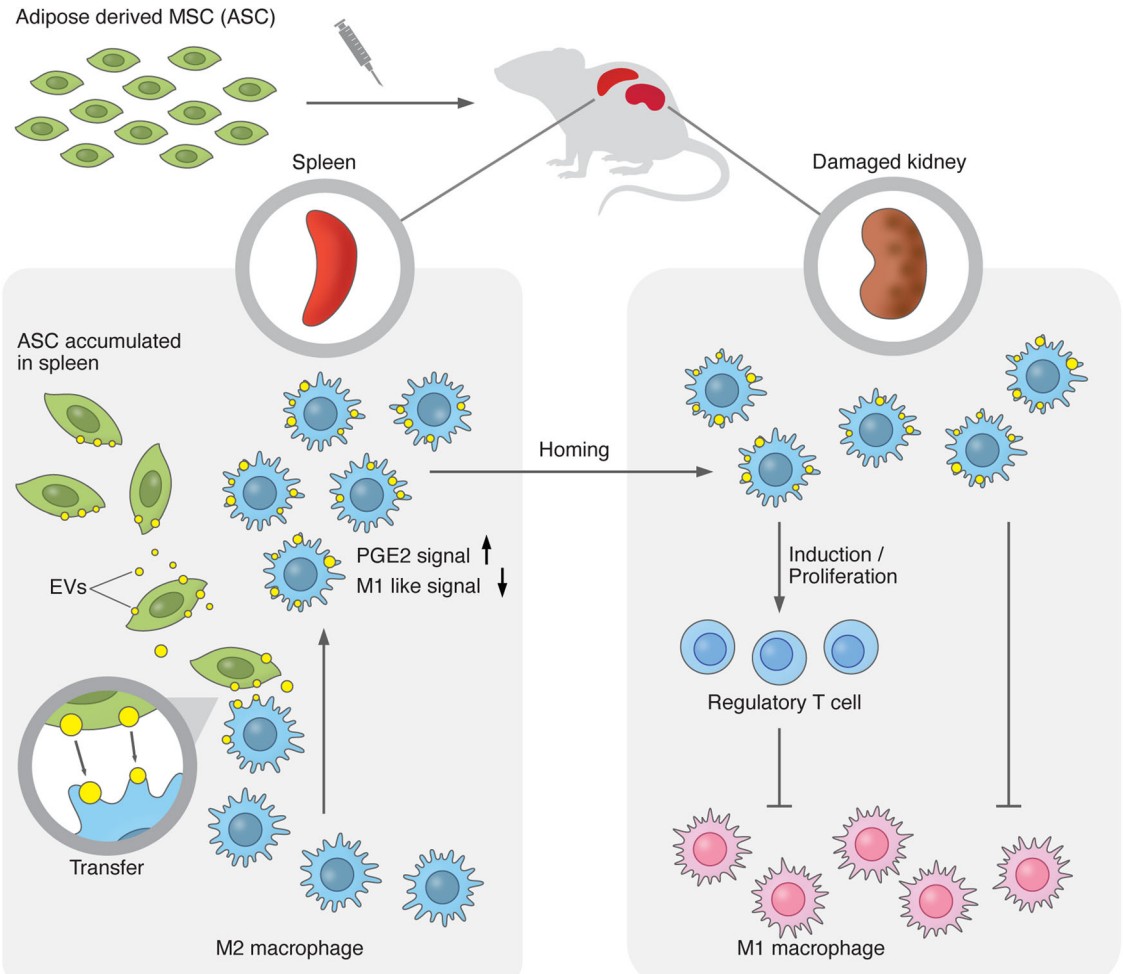

**Fig. 9 Scheme of the proposed action of ASCs and ASC-derived EVs in nephritis.** When ASCs are administered to rats with nephritis, ASC-derived EVs are transferred to M2 macrophages in the spleen, leading to macrophage activation via PGE2 stimulation and reduction of M1 signatures. EV-transferred M2 macrophages move from the spleen to the inflamed kidney, resulting in the increase of Treg and decrease of M1 macrophages. ASCs exert renoprotection via EV-mediated modulation of M2 macrophages and the spleen-kidney network.

(Sigma-Aldrich, St. Louis, MO) and transferred to culture plates. After overnight incubation at 37 °C in a 5% CO$_2$ atmosphere, floating cells were removed from the culture plates. Then, DiD-labeled ASCs were added to the macrophage culture plate at a 1:50 ratio. Fluorescence and phase-contrast images were captured every 5.5 min using a BZ-X800 incubator microscope system (Keyence, Tokyo, Japan) and analyzed with a BZ-X800 analyzer (Keyence, Tokyo, Japan).

**In vivo dynamic imaging of DiD-positive leukocytes in the spleen.** The dynamics of DiD-positive leukocytes were observed using an intravital 2-photon microscope. A female 3-week-old WKY rat was injected with 50 μg TF78 and DiD-labeled ASCs ($1 \times 10^6$ cells) on day 0. On day 2, the rat was anesthetized with a mixture of medetomidine midazolam and butorphanol and injected with 0.25 μg/g body weight FITC anti-rat CD45 (Biolegend, 202226) and 0.01 mg/g body weight rhodamine-70k dextran. Then, the spleen was exteriorized with a lateral incision and extended over the imaging platform. Time-lapse images were acquired at 30 s per frame using a Nikon A1RMP microscope.

**Co-culture of CD4$^+$ T cells with ASCs, ASC-derived EVs, and EVs-transferred macrophages.** Bone marrow cells or CD4$^+$ T cells were obtained from the tibias and femurs of 9–18-week-old normal WKY rats. Bone marrow fragments were passed by ice-cold PBS with a 22 G needle and a 10 mL syringe, washed, centrifuged, and treated with red blood cell lysis buffer. Magnetic cell separation was performed to harvest CD4-positive cell-rich samples using an EasySep rat CD4$^+$ T-cell isolation kit (STEMCELL Technologies, Vancouver, BC, Canada). After this negative selection procedure, samples were stained with fluorophore-conjugated antibodies listed in Supplementary Table 3G, and live CD4$^+$ T cells were sorted using FACS SORP Aria II. Bone marrow cells and ASCs were co-cultured at a 5:1 ratio in RPMI 1640 medium (Sigma-Aldrich, R8758) with 10% FBS and 1%

penicillin-streptomycin (Gibco, 1570-063). CD4$^+$ T cells and ASCs were co-cultured in a 1:1 ratio. CD4$^+$ T cells and EV$^+$ or EV$^-$ macrophages were co-cultured at a 5:1 ratio. For co-culture with EVs, EVs were added to the RPMI 1640 medium at a 41 μg/mL concentration.

**Observation of the lungs using IVIS imaging system.** ASCs were labeled with 5 μg/mL XenoLight DiR-dye (Perkin Elmer, MA, USA, 12594) in PBS and administrated with or without heparin. The lungs were harvested for ex vivo imaging using an IVIS Spectrum CT system (Caliper Life Sciences, MA, USA) on day 3. The filter set for DiR-dye imaging was 745 nm excitation and 800 nm emission.

**Statistics and reproducibility.** All experiments were repeated independently, and statistical methods are described in the figure legends. Sample sizes were chosen empirically based on our preliminary experiments performed using the animal model to ensure adequate statistical power. To evaluate statistical differences among the three groups, $p$ values were calculated by ordinary one-way analysis of variance followed by Tukey's multiple comparison tests. The Welch's $t$ test was used to evaluate the difference between the two groups. All statistical analyses were performed using GraphPad Prism (version9, GraphPad Software, LLC).

**Reporting summary.** Further information on research design is available in the Nature Research Reporting Summary linked to this article.

## Data availability
Reasonable requests for additional data or materials will be fulfilled under appropriate agreements. RNA sequencing data are available in the Gene Expression Omnibus under

accession number GSE179301. All data generated or analyzed in this study are included in this published article. The source data underlying most graphs and charts used in this manuscript are provided as a Supplementary Data File. Request for any source data or materials that are not provided should be made to the corresponding author.

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

## Acknowledgements

We thank Noriyuki Suzuki, Naoko Asano, and Ayako Sakamoto for their excellent assistance. We would like to thank Editage (www.editage.com) for English language editing. This study was supported by the Aichi Kidney Foundation (Y.S.), JSPS KAKENHI under Grant Number JP19K08722 (K.F.), JST FOREST Program under Grant Number JPMJFR200W (K.F.), AMED under Grant Number JP21bm0404075 (K.F.), and JSPS KAKENHI Home-Returning Researcher Development Research Grant under Grant Number JP19K24694 (H.I.S.).

## Author contributions

K.F. and S. Maruyama conceived and designed this project. Y. Shimamura, K.F., A.T., M.K., T.N., S.K., K.W., A.S., S. Minatoguchi, M.M., Y. Sawa, N.T., T.I., H.I.S., and S. Maruyama designed and performed the experiments and analyzed the data. Y. Shimamura, K.F., H.I.S. interpret data and wrote the manuscript. K.F., H.I.S., and S. Maruyama supervised this interdisciplinary team and proofread the manuscript. All co-authors approved the manuscript.

## Competing interests

The authors declare no competing interests.
