## [Peer Review File · Communications Biology]

Reviewers' comments:

Reviewer #1 (Remarks to the Author):

In this study, Shimamura, Furuhashi, Tanaka, et al. uncover a mechanism of immunomodulation alleviating renal damage in rats by transferring human stem cells. The anti-inflammatory effect of adipo/mesenchymal cells is well established and drives numerous efforts to develop stem cell-based treatments. The significance of this study is in demonstrating a mechanism of immunomodulation that acts through macrophages and T cells. Additionally to the kidney, the ASC components were probed in the liver, lungs, and spleen. This led to finding a significant enrichment in the spleen and that immunomodulation of macrophages and T cells in the spleen is parallel to the observed effect in the kidney. Moreover, spleen immunomodulation was found to be necessary for kidney immunomodulation resulting in reduced kidney inflammation and damage. The study includes a comprehensive flow cytometry data analysis of leukocytes presented in an informative way. Furthermore, RNAseq analysis explores the effect of EV treatment on macrophages. A key effect is demonstrated on macrophages – being a central orchestrator of immunity and highly active in the uptake and processing of cellular components. These cells are likely to uptake the EV and respond to their cargo. Importantly, this study demonstrated Tregs induction in whole BM or CD4+ cells by ASC, ASC derived EV, and EV treated M2 macrophages. Major:

1. Improve the validity of the definitions of cell population in flow cytometry data. This study doesn't mention monocytes. The original study that proposed the gating strategy for flow cytometry data was termed CD43Lo/His48Hi and CD43Hi/His48Int-Lo monocyte-macrophages. These were analyzed in the liver, lungs, spleen, and BM. Is there a specific circumstance in the kidney for not sticking to the original labels for this gating strategy? M1 and M2 macrophages are a convenient simplification that is necessary to communicate data. However, I think that in this case ignoring monocytes and not adhering to the labels proposed in the original study creates confusion. Please provide a rationale for changing the population names or adjust accordingly.

2. Add data. Although the flow analysis is comprehensive regarding leukocyte populations in CD45+, leukocyte numbers in tissues are missing. The authors refer to a previous work where a histological assessment of leukocyte infiltration is presented for days 1,3,7 and 14. However, assuming that similar portions of organs were processed for flow cytometry, I suggest presenting the portion of leukocytes in the total analyzed cells. Also, the numbers can be normalized to tissue mass. Similar to normalization to tissue area in histology.

3. Clarify the distinguishment between transferred and phagocytized EV. If possible, probe for relevant data from DEG. The authors attempt to distinguish between DiD labeled EV transfer and phagocytosis. It is not clear what will be the biological meaning of attempting to distinguish transfer and phagocytosis when studying macrophages. It is well established that MSC deliver EVs. Also, it is well established that macrophages are highly phagocytic and especially under an M2-like profile. Phagocytosis of any cargo involves several sequential steps of recognition, binding, uptake, and catalytic processing. Therefore, concluding microscopic snapshots of particles on CD45+ is confusing. The stage of uptake is not clear and previously digested cargo might not give a signal. Of note, although the video clip of EV transmission clearly illustrates the process, it cannot tell fine structures such as macrophage extensions that catch the EV. I wonder if the DEG analysis might provide additional data regarding phagocytosis-related pathways and genes.

4. The manuscript proposes EV transfer is specific to M2 macrophages. However, this is not tested directly. Previous work of the authors did demonstrate EV induction of immunoregulatory / M2 macrophages. This corresponds to an established effect of MSC EV macrophage reprogramming through several mechanisms that induce mostly M2-like features. E.g:

<https://doi.org/10.3389/fimmu.2018.00771>

<https://doi.org/10.1002/stem.2372>

<https://doi.org/10.1164/rccm.201701-0170OC>

Therefore, it is not completely clear whether the observations in this manuscript derive from EV-induced M2, EV-specific delivery to M2, or both. Please clarify your mechanism and conclusion regarding these- EV promote M2 specifically / EV accumulate in M2 specifically / M2 specifically uptake EV / etc.

5. RNAseq

5.a. DEG. Color coding in Fig 6 b and c don't match which makes it difficult to follow.

Fig 6 c headline is not clear. Are EV + mean EV + and - ?

5. b. This analysis shows mostly that EV treatment produces an effect on macrophages. Regarding EV – and +, the authors suggest some trends in the DEG data between EV + and -.

The data is presented in Z score means without any specific genes. Within each group, genes that drive the statistical parameters might be more or less relevant. Could you add specific DEG genes in EV negative vs. EV positive that represent the effect?

5. c. RNAseq GO. The authors write that 'functions of secretion, exocytosis, glycolysis, and myeloid leukocyte activation, suggesting activation of M2 macrophages by EVs'. Unfortunately, It is not clear why. e.g glycolysis is found many times in inflammatory (M1) macrophages. Providing references that support these suggestions and conclusions will help to understand them. Also, GO terms might be generic. Are there specific genes within these GO terms that have an established involvement and might represent the effect?

5.d. Also, please explain the GO presentation. Are these terms high in a statistical score? How many genes are in each?

5.e. RNAseq GO. The authors write in line 194 ' genes in DEG groups 4, 5, and 6 were downregulated by nephritis, and the expression of these genes was further suppressed in EV+ samples (Fig. 6c)'.

Group 5 genes in the figure show a lower decrease in EV treatment.

Also, 'These genes were enriched for IFN- γ , TNF- α , and 195 NF- κ B pathways, which are important for the induction of M1 macrophages'. This is indeed important, however, are there any specific genes with significant downregulation? Any representatives?

5. f. What are the genes/pathways in group 7? Any explanation?

Thank you,

Reviewer #2 (Remarks to the Author):

The manuscript entitled, " Mesenchymal stem cells exert renoprotection via extracellular vesicle-mediated modulation of M2 macrophages and spleen-kidney network" by Shimamura, et al., attempts to address the mechanism by which adipose-derived mesenchymal stem cells (ASCs) can serve as a therapeutic for nephritis. By injecting ASCs into a glomerulonephritis model and comparing therapeutic ASC effects to bone-marrow-derived mesenchymal stem cells (BMMSCs) it was found that ASCs preferentially affected nephritis outcomes more so than BMMSCs. This therapeutic effect was due to the transition of M2 macrophages, which did not occur with BMMSC treatment. Though the model is one for nephritis, very few of the injected cells migrated to the kidney, and most were enriched in the spleen. The therapeutic effects were ablated when the spleen was removed, suggesting the spleen plays an important role. Interestingly, the group reported a finding that the ASCs were secreting extracellular vesicles (EVs) which helped the splenic M2 macrophage conversion. They then examined the gene expression profiles of the M2 macrophages affected by the ASC EVs. Further, they found that the ASC-derived EVs themselves could affect nephritis through the induction of Tregs.

This manuscript is very well presented, very well written, and the data are well analyzed. Some fundamental issues need to be addressed. The authors' stated goal was to determine the mechanism by which ASCs could therapeutically benefit nephritic disease state. However, it seems the data add to the phenomenon without directly addressing the mechanism. Importantly, this was displayed by the splenectomy, which ablated the ASC therapeutic effects. Therefore, there is a signal within the spleen that is causing ASCs to secrete EVs and a signal within the EVs that affects macrophage polarization. The polarized M2 cells can then home to the kidney and induce Tregs to dampen the inflammatory response within the kidney leading to a beneficial outcome. The mechanism, therefore, lies within the spleen and the EVs.

1.) What signal is being produced by the spleen that is causing ASC EV secretion?

2.) What signals within the EVs are causing the M2 macrophage polarization?

These two questions will address the mechanism. A transcriptomic profile of the spleen upon ASC injection, when compared to BMMSC injection, will determine which specific splenic pathways are activated leading to ASC activation. The activation could be cell-cell contact or secretion of a key molecule by a splenic cell. This would be a good mechanism. Further, the research group used two

different methods to identify and purify EVs, flow cytometry and ultracentrifugation. Use either of these methods to purify the ASC-derived EVs such that they can be analyzed by mass spectrometry to determine the contents of the EVs. The molecules within the EVs are driving the M2 polarization, if those molecules are identified, the injection of the ASCs themselves may not be necessary.

Admittedly, while writing this review it has become apparent that asking for these data to solve the mechanism may be more than this manuscript needs to address. There is a lot of good data in this study that needs to be shown to the scientific community. However, it is not addressing the mechanism directly, it is continuing to elucidate the phenomenon. This quality manuscript should be accepted nearly as is, so long as the authors refrain from using the term "mechanism" throughout the manuscript to describe the effects of ASCs on nephritis. The mechanism is still undetermined.

Point by point responses to the reviewers' comments.

Responses to Reviewer #1

Reviewers' comments:

Reviewer #1 (Remarks to the Author):

In this study, Shimamura, Furuhashi, Tanaka, et al. uncover a mechanism of immunomodulation alleviating renal damage in rats by transferring human stem cells. The anti-inflammatory effect of adipo/mesenchymal cells is well established and drives numerous efforts to develop stem cell-based treatments. The significance of this study is in demonstrating a mechanism of immunomodulation that acts through macrophages and T cells. Additionally to the kidney, the ASC components were probed in the liver, lungs, and spleen. This led to finding a significant enrichment in the spleen and that immunomodulation of macrophages and T cells in the spleen is parallel to the observed effect in the kidney. Moreover, spleen immunomodulation was found to be necessary for kidney immunomodulation resulting in reduced kidney inflammation and damage. The study includes a comprehensive flow cytometry data analysis of leukocytes presented in an informative way. Furthermore, RNAseq analysis explores the effect of EV treatment on macrophages. A key effect is demonstrated on macrophages – being a central orchestrator of immunity and highly active in the uptake and processing of cellular components. These cells are likely to uptake the EV and respond to their cargo. Importantly, this study demonstrated Tregs induction in whole BM or CD4+ cells by ASC, ASC derived EV, and EV treated M2 macrophages.

Major:

Reviewer comments	Author replies
1. Improve the validity of the definitions of cell population in flow cytometry data. This study doesn't mention monocytes. The original study that proposed the gating strategy for flow cytometry data was termed CD43Lo/His48Hi and CD43Hi/His48Int-Lo monocyte-macrophages. These were analyzed in the liver, lungs, spleen, and BM. Is there a specific circumstance in the kidney for not sticking to the original labels for this gating strategy? M1 and M2 macrophages	In this study, we analyzed the leukocytes in the kidneys after eliminating all peripheral blood by blood withdrawal. Therefore, circulating monocytes were not included in the present analysis. In accordance with your suggestion, we have added this detail to the METHODS. As you pointed out, macrophages are diverse and can be broadly classified into inflammatory macrophages and anti-inflammatory macrophages. In light of this major classification, we referred to inflammatory macrophages as M1 macrophages and

are a convenient simplification that is necessary to communicate data. However, I think that in this case ignoring monocytes and not adhering to the labels proposed in the original study creates confusion. Please provide a rationale for changing the population names or adjust accordingly.	anti-inflammatory macrophages as M2 macrophages as in this study. Considering M2 macrophage diversity, RNA-seq analysis was used to compare M2 macrophages with changes induced by 28 stimulating factors. In general, it would be easier for readers to understand if the cell population that is being focused on is described as M2 macrophages, which are anti-inflammatory macrophages. However, following your suggestion, we have stated in the text that there is diversity among M2 macrophages.
2. Add data. Although the flow analysis is comprehensive regarding leukocyte populations in CD45+, leukocyte numbers in tissues are missing. The authors refer to a previous work where a histological assessment of leukocyte infiltration is presented for days 1,3,7 and 14. However, assuming that similar portions of organs were processed for flow cytometry, I suggest presenting the portion of leukocytes in the total analyzed cells. Also, the numbers can be normalized to tissue mass. Similar to normalization to tissue area in histology.	Thank you for pointing this out. Although we also used counting beads to obtain cell count information, we were unable to accurately evaluate the cell count because the beads were attached to the kidney tissue fragments and the cell count was drastically reduced during the staining process due to the weakened tubular cells in nephritis. You suggested an appropriate alternative measure that is close to cell count, that is, CD45 cell percentage of live cells. Therefore, we followed your suggestion and have added the data on CD45 cell percentage of live cells to Supplementary Figure 3.

3. Clarify the distinction between transferred and phagocytized EV. If possible, probe for relevant data from DEG. The authors attempt to distinguish between DiD labeled EV transfer and phagocytosis. It is not clear what will be the biological meaning of attempting to distinguish transfer and phagocytosis when studying macrophages. It is well established that MSC deliver EVs. Also, it is well established that macrophages are highly phagocytic and especially under an M2-like profile. Phagocytosis of any cargo involves several sequential steps of recognition, binding, uptake, and catalytic processing. Therefore, concluding microscopic snapshots of particles on CD45+ is confusing. The stage of uptake is not clear and previously digested cargo might not give a signal. Of note, although the video clip of EV transmission clearly illustrates the process, it cannot tell fine structures such as macrophage extensions that catch the EV. I

3. Clarify the distinction between transferred and phagocytized EV. If possible, probe for relevant data from DEG. The authors attempt to distinguish between DiD labeled EV transfer and phagocytosis. It is not clear what will be the biological meaning of attempting to distinguish transfer and phagocytosis when studying macrophages. It is well established that MSC deliver EVs. Also, it is well established that macrophages are highly phagocytic and especially under an M2-like profile. Phagocytosis of any cargo involves several sequential steps of recognition, binding, uptake, and catalytic processing. Therefore, concluding microscopic snapshots of particles on CD45+ is confusing. The stage of uptake is not clear and previously digested cargo might not give a signal. Of note, although the video clip of EV transmission clearly illustrates the process, it cannot tell fine structures such as macrophage extensions that catch the EV. I

We performed high-resolution imaging to determine whether MSC-derived cell membrane components were transferred as EVs or whether apoptotic MSCs were phagocytosed. CD45+DiD+ cells, in which DiD particles occupied more than 1/3 of the cytoplasm area, were considered as leukocytes phagocytosing ASCs. CD45+ cells with DiD particles on the cell membranes were identified as EV-transferred leukocytes. As per your suggestion, we have revised the description of the figure for clarity purposes (Fig. 5c).

e

ASC, Apoptotic ASC, EV

CD45/DiD/DAPI

Percentage in CD45+DiD+ cells: Spleen, Kidney

Legend: ASC (red), Apoptotic ASC (yellow), EV (green)

According to your suggestion, we analyzed the relationships between DEG groups and gene sets related to phagocytosis. However, we did not observe a consistent enrichment of phagocytosis-related gene sets in either the upregulated or the downregulated genes. Therefore, with respect to gene expression profiling, the modulation of phagocytosis is unclear in our datasets.

wonder if the DEG analysis might provide additional data regarding phagocytosis-related pathways and genes.	
4. The manuscript proposes EV transfer is specific to M2 macrophages. However, this is not tested directly. Previous work of the authors did demonstrate EV induction of immunoregulatory / M2 macrophages. This corresponds to an established effect of MSC EV macrophage reprogramming through several mechanisms that induce mostly M2-like features. E.g: https://doi.org/10.3389/fimmu.2018.00771 https://doi.org/10.1002/stem.2372 https://doi.org/10.1164/rccm.201701-01700C Therefore, it is not completely clear whether the observations in this manuscript derive from EV-induced M2, EV-specific delivery to M2, or both. Please clarify your mechanism and conclusion regarding these- EV promote M2 specifically / EV accumulate in M2 specifically / M2 specifically uptake EV / etc.	The word "specifically" has been changed to "predominantly" or "mainly" to avoid misunderstanding. As you indicated, we performed a flow cytometry assessment of EVs-positive cells at earlier time points to accurately confirm that the EVs were transferred to the M2 macrophages. Even as early as 4 hours after ASCs administration, the majority of EVs were found in M2 macrophages, suggesting that ASCs-derived EVs were predominantly transferred to M2 macrophages (Supplementary Fig. 6). RNA-seq analysis was performed by sorting M2 macrophages. In this RNA-seq analysis, EVs-positive M2 macrophages showed enhanced anti-inflammatory functional changes, suggesting that EVs may induce hyperpolarization in M2 macrophages. Therefore, EVs delivered to M2 macrophages induced hyperpolarization of M2 macrophages as well. We have updated the relevant description in the revised manuscript. Supplementary figure 6 5. RNAseq 5.a. DEG. Color coding in Fig 6 b and c don't match which makes it difficult to follow. Fig 6 c headline is not clear. Are EV + mean EV + and - ?	We appreciate your careful reading of our manuscript. According to your suggestion, we have revised the color codes in Fig. 6 in the revised manuscript. In addition, the headlines in Fig. 6d were corrected from "GN&EV(+)" to "GN&EV(+/-)".

5. b. This analysis shows mostly that EV treatment produces an effect on macrophages. Regarding EV – and +, the authors suggest some trends in the DEG data between EV + and -. The data is presented in Z score means without any specific genes. Within each group, genes that drive the statistical parameters might be more or less relevant. Could you add specific DEG genes in EV negative vs. EV positive that represent the effect?	According to your suggestion, we have included a heatmap of the representative DEG genes in Fig. 6c in the revised manuscript. These genes are associated with “secretion, exocytosis, glycolysis, and myeloid leukocyte activation” and “IFN-γ, TNF-α, and NF-κB pathways” and were described in the following responses.
5. c. RNAseq GO. The authors write that ‘functions of secretion, exocytosis, glycolysis, and myeloid leukocyte activation, suggesting activation of M2 macrophages by EVs’. Unfortunately, It is not clear why. e.g glycolysis is found many times in inflammatory (M1) macrophages. Providing references that support these suggestions and conclusions will help to understand them. Also, GO terms might be generic. Are there specific genes within these GO terms that have an established involvement and might	We appreciate this important comment. First, a recent report (PMID: 34133934, Cell Rep. 2021 Jun 15;35(11):109246. doi: 10.1016/j.celrep.2021.109246.) has shown that the stimulation of M2 macrophage with succinate induces hyperpolarization of M2 macrophages, which is associated with characteristic transcriptome changes including upregulation of genes involved in secretion and exocytosis pathways and downregulation of genes that are preferentially expressed in M1 macrophages. Transcriptome signatures of succinate-induced M2 hyperpolarization are similar to those of our datasets.

represent the effect?

Second, as you suggested, it is generally accepted that M1 macrophages rely mainly on glycolysis, whereas M2 macrophages are more dependent on mitochondrial OXPHOS. However, recent studies have suggested that macrophage metabolism is not as simple as presumed previously and that glycolysis is also important for M2 macrophages (PMID: 33407885, Biomark Res. 2021 Jan 6;9(1):1. doi: 10.1186/s40364-020-00251-y.). We have included a portion of this description in the main text of the revised manuscript. In addition, we have included a heatmap of the representative DEG genes in Fig. 6c in the revised manuscript. Representative genes associated with “secretion, exocytosis, glycolysis, and myeloid leukocyte activation” include *Plod1*, *Gusb*, *Chst1* (glycolysis), *Hmgcr*, *Slc12a2*, *Pcsk1*, *Llg12*, *Anxa3*, *Itgam*, *Fcgr2b*, and *Tnfaip2* (secretion and exocytosis). These findings collectively suggest that M2 macrophages undergo hyperpolarization in nephritis and that EVs mediated a further phenotypic shift probably toward anti-inflammatory phenotypes. We have updated the relevant description in the revised manuscript accordingly.

5.d. Also, please explain the GO

In Fig. 6d in the revised manuscript, FDR q-values are

presentation. Are these terms high in a statistical score? How many genes are in each?	displayed. In addition, we have included a table showing the number of genes in each gene set and the overlapped genes in Supplementary Table X3.
5.e. RNAseq GO. The authors write in line 194 ‘ genes in DEG groups 4, 5, and 6 were downregulated by nephritis, and the expression of these genes was further suppressed in EV+ samples (Fig. 6c)’. Group 5 genes in the figure show a lower decrease in EV treatment.	We have modified the relevant sentences as follows: “On the other hand, genes in DEG groups 4, 5, and 6 showed downregulation by nephritis, and the expression of genes in DEG groups 4 and 6 were further suppressed in EV+ samples (Fig. 6c)”.
Also, ‘These genes were enriched for IFN-γ, TNF-α, and 195 NF-κB pathways, which are important for the induction of M1 macrophages. This is indeed important, however, are there any specific genes with significant downregulation? Any representatives?’	We have included a heatmap of the representative DEG genes in Fig. 6c in the revised manuscript. The representative genes associated with “IFN-γ, TNF-α, and NF-κB pathways” include Helz2, Cxcl10, Cxcl11, Tnfaip2, Ccl2, Icam1, Pim1, Nr4a3, Hes1, Fosb, Phlda1, Zfp36, Btg2, and Junb. We appreciate the important comments from the reviewers. We have updated the relevant description in the revised manuscript.
5. f. What are the genes/pathways in group 7? Any explanation? Thank you,	DEG group 7 includes granzyme b and probably reflects contamination with NK cells or T cells, especially in one sample of EV(-) group. Therefore, we did not focus on this DEG group. We have updated the relevant description in the revised manuscript.

Revised parts of the text are highlighted in yellow.

Responses to Reviewer #2

Reviewer #2 (Remarks to the Author):

The manuscript entitled, “ Mesenchymal stem cells exert renoprotection via extracellular vesicle-mediated modulation of M2 macrophages and spleen-kidney network” by Shimamura, et al., attempts to address the mechanism by which adipose-derived mesenchymal stem cells (ASCs) can serve as a therapeutic for nephritis. By injecting ASCs into a glomerulonephritis model and comparing therapeutic ASC effects to bone-marrow-derived mesenchymal stem cells (BMMSCs) it was found that ASCs preferentially affected nephritis outcomes more so than BMMSCs. This therapeutic effect was due to the transition of M2 macrophages, which did not occur with BMMSC treatment. Though the model is one for nephritis, very few of the injected cells migrated to the kidney, and most were enriched in the spleen. The therapeutic effects were ablated when the spleen was removed, suggesting the spleen plays an important role. Interestingly, the group reported a finding that the ASCs were secreting extracellular vesicles (EVs) which helped the splenic M2 macrophage conversion. They then examined the gene expression profiles of the M2 macrophages affected by the ASC EVs. Further, they found that the ASC-derived EVs themselves could affect nephritis through the induction of Tregs.

This manuscript is very well presented, very well written, and the data are well analyzed. Some fundamental issues need to be addressed. The authors’ stated goal was to determine the mechanism by which ASCs could therapeutically benefit nephritic disease state. However, it seems the data add to the phenomenon without directly addressing the mechanism. Importantly, this was displayed by the splenectomy, which ablated the ASC therapeutic effects. Therefore, there is a signal within the spleen that is causing ASCs to secrete EVs and a signal within the EVs that affects macrophage polarization. The polarized M2 cells can then home to the kidney and induce Tregs to dampen the inflammatory response within the kidney leading to a beneficial outcome. The mechanism, therefore, lies within the spleen and the EVs.

- 1.) What signal is being produced by the spleen that is causing ASC EV secretion?
- 2.) What signals within the EVs are causing the M2 macrophage polarization?

These two questions will address the mechanism. A transcriptomic profile of the spleen upon ASC injection, when compared to BMMSC injection, will determine which specific splenic pathways are activated leading to ASC activation. The activation could be cell-cell contact or secretion of a key molecule by a splenic cell. This would be a good mechanism. Further, the research group used two different methods to identify and purify EVs, flow cytometry and ultracentrifugation. Use either of these methods to purify the ASC-derived EVs such that they can be analyzed by mass spectrometry to

determine the contents of the EVs. The molecules within the EVs are driving the M2 polarization, if those molecules are identified, the injection of the ASCs themselves may not be necessary.

Admittedly, while writing this review it has become apparent that asking for these data to solve the mechanism may be more than this manuscript needs to address. There is a lot of good data in this study that needs to be shown to the scientific community. However, it is not addressing the mechanism directly, it is continuing to elucidate the phenomenon. This quality manuscript should be accepted nearly as is, so long as the authors refrain from using the term “mechanism” throughout the manuscript to describe the effects of ASCs on nephritis. The mechanism is still undetermined.

Reviewer comments	Author replies
1.) What signal is being produced by the spleen that is causing ASC EV secretion?	MSCs are known to produce EVs, and many investigations have attempted to concentrate MSC-derived EVs to improve injured organs with EVs alone. However, even in our results in this study, the therapeutic effect of EVs obtained from culture is weaker than that of the cells themselves, suggesting that inflammatory conditions in vivo may modulate MSC-derived EVs or that the amount of EVs obtained is insufficient. While the mechanism of production of MSC-derived EVs has not been well understood, there have been several reports on the factors that affect the function of EVs. It has been reported that inflammatory stimulation with interferon gamma enhances the anti-inflammatory function of MSCs or MSC-derived EVs. To examine whether the EVs secreted in vitro are the same as those secreted in vivo after inflammatory stimuli, further studies are needed.

2.) What signals within the EVs are causing the M2 macrophage polarization?	As you pointed out, it is important to identify the factors of EVs released by MSCs administered in vivo that act on macrophages in order to investigate the therapeutic potential of MSCs. We successfully tracked the MSCs-derived EVs in vivo by labeling the plasma membrane of administered MSCs. The kidneys contained very few of the administered MSCs themselves, mostly leukocytes to which the MSC-derived EVs were transferred. Furthermore, the majority of leukocytes transferred with EVs were M2 macrophages. Thus, M2 macrophages to which MSCs-derived EVs are transferred in vivo can be detected by flow cytometry. In this study, M2 macrophages to which EVs were transferred were isolated using flow cytometry, and functional changes in EVs-transferred M2 macrophages were analyzed using RNA-seq. To the best of our knowledge, this is the first transcriptome study to analyze the phenotypic changes in M2 macrophages induced by EVs secreted in vivo by MSCs. Considering the diversity of M2 macrophages, we investigated for the EVs stimuli that were comparable to the 28 stimuli by comparing the genetic changes in M2 macrophages induced by the 28 stimuli. The results suggested that EV transfer facilitates hyperpolarization of M2 macrophages in nephritis conditions further in the M2 direction possibly via PGE2 stimulation. This has been included in the discussion section
--	--

3.) Admittedly, while writing this review it has become apparent that asking for these data to solve the mechanism may be more than this manuscript needs to address. There is a lot of good data in this study that needs to be shown to the scientific community. However, it is not addressing the mechanism directly, it is continuing to elucidate the phenomenon. This quality manuscript should be accepted nearly as is, so long as the authors refrain from using the term “mechanism” throughout the manuscript to describe the effects of ASCs on nephritis. The mechanism is still undetermined.	Thank you for this important suggestion. We have replaced the word "mechanism" with the words "effect", "action" or "phenomenon". We have updated the relevant description in the revised manuscript.
---	---

Revised parts of the text are highlighted in yellow.

REVIEWERS' COMMENTS:

Reviewer #1 (Remarks to the Author):

The authors responded satisfactorily to all review sections.
I am sure that this work will be of interest to researchers in the field.
Thanks you,

Reviewer #2 (Remarks to the Author):

The authors have made the necessary minor changes to the manuscript needed for publication.

Point by point responses to the reviewers' comments.

Responses to Reviewer #1

Reviewers' comments:

Reviewer #1 (Remarks to the Author):

Reviewer comments	Author replies
The authors responded satisfactorily to all review sections. I am sure that this work will be of interest to researchers in the field. Thanks you,	Thank you.

Responses to Reviewer #2

Reviewer comments	Author replies
The authors have made the necessary minor changes to the manuscript needed for publication.	Thank you.